# Mutant GGGGCC RNA prevents YY1 from binding to *Fuzzy* promoter which stimulates Wnt/β-catenin pathway in C9ALS/FTD

Zhefan Stephen Chen [1,2], Mingxi Ou[1], Stephanie Taylor [2], Ruxandra Dafinca [2,3], Shaohong Isaac Peng[1], Kevin Talbot [2,3] ✉ & Ho Yin Edwin Chan [1,4] ✉

The GGGGCC hexanucleotide repeat expansion mutation in the chromosome 9 open reading frame 72 (*C9orf72*) gene is a major genetic cause of amyotrophic lateral sclerosis and frontotemporal dementia (C9ALS/FTD). In this study, we demonstrate that the zinc finger (ZF) transcriptional regulator Yin Yang 1 (YY1) binds to the promoter region of the planar cell polarity gene *Fuzzy* to regulate its transcription. We show that YY1 interacts with GGGGCC repeat RNA via its ZF and that this interaction compromises the binding of YY1 to the *Fuzzy*[YY1] promoter sites, resulting in the downregulation of *Fuzzy* transcription. The decrease in Fuzzy protein expression in turn activates the canonical Wnt/β-catenin pathway and induces synaptic deficits in C9ALS/FTD neurons. Our findings demonstrate a *C9orf72* GGGGCC RNA-initiated perturbation of YY1–*Fuzzy* transcriptional control that implicates aberrant Wnt/β-catenin signalling in C9ALS/FTD-associated neurodegeneration. This pathogenic cascade provides a potential new target for disease-modifying therapy.

The non-coding hexanucleotide GGGGCC repeat expansion mutation in the chromosome 9 open reading frame 72 (*C9orf72*) gene is a major genetic cause of amyotrophic lateral sclerosis (ALS), a progressive and fatal neurodegenerative disorder characterised by a loss of motor neurons and paralysis[1,2]. Frontotemporal dementia (FTD), in which the loss of prefrontal cortical neurons and their connections leads to behavioural change, shares genetic and pathological features with ALS, exemplified by the fact that mutation in the *C9orf72* gene underlies a significant proportion of FTD cases[3]. C9ALS/FTD has been used to collectively refer to patients with clinical features of ALS and/or FTD. Three non-exclusive pathogenic mechanisms have been implicated in C9ALS/FTD: (1) RNA-mediated toxicity[4]; (2) repeat-associated non-ATG translation of dipeptide proteins[5] produced by both sense and anti-sense strands of the pathogenic allele, which induce neuronal dysfunction by interacting with key cellular proteins; and (3) loss of endogenous C9orf72 protein function due to reduced *C9orf72*

transcription resulting from the GGGGCC repeat expansion (*C9orf72* haploinsufficiency)[6]. Elucidating the effects of GGGGCC repeat expansion on molecular pathways is essential for a better understanding of the disease mechanisms underlying C9ALS/FTD.

Yin Yang 1 (YY1) is a versatile transcriptional regulator that ensures proper transcriptional profiling in neurons[7]. In individuals with neurodegenerative disorders, the function of YY1 is compromised as it is recruited to disease protein aggregates[8], mislocalised within the cell[9] and/or proteolytically degraded[10,11]. Currently, the role of YY1 in C9ALS/FTD pathogenesis is unclear. Yin et al.[9] reported that when compared with the astrocytes isolated from a mouse model overexpressing the wild-type form of human superoxide dismutase 1 (SOD1), the concentration of nuclear YY1 protein was increased in the astrocytes carrying an ALS-causing SOD1[G93A] mutation. The accumulation of nuclear YY1 attenuates the expression of excitatory amino acid transporter-2, a protein that mediates glutamate uptake from the

[1]School of Life Sciences, Faculty of Science, The Chinese University of Hong Kong, Shatin, N.T., Hong Kong SAR, China. [2]Oxford Motor Neuron Disease Centre, Nuffield Department of Clinical Neurosciences, John Radcliffe Hospital, University of Oxford, Oxford OX3 9DU, UK. [3]Kavli Institute for Nanoscience Discovery, University of Oxford, Dorothy Crowfoot Hodgkin Building, South Parks Road, Oxford OX1 3QU, UK. [4]Gerald Choa Neuroscience Institute, The Chinese University of Hong Kong, Shatin, N.T., Hong Kong SAR, China. ✉e-mail: kevin.talbot@ndcn.ox.ac.uk; hyechan@cuhk.edu.hk

extracellular compartment[12]. This attenuation in turn leads to insufficient glutamate clearance and neuronal excitotoxicity[9].

The Fuzzy protein is classified as one of the ciliogenesis and planar polarity effector (CPLANE) proteins, which comprise additional members, including Inturned, Wdpcp, Rsg1 and Jbts17[13]. Different CPLANE members control protein transport into, along and out of ciliary axonemes in support of ciliary outgrowth in vertebrate cells[13]. Cell migration defects that lead to developmental neural tube closure defects have been observed in *Fuzzy*-null (*Fuzzy*$^{-/-}$) mutant mice[14]. In humans, mutations in *Fuzzy* are known to cause neural tube defects[15]. Recently, Fuzzy has been implicated in human neurodegenerative disorders[8] and cancer[16–18]. We previously reported that YY1 is recruited to polyglutamine aggregates and amyloid beta peptide 42 (Aβ$_{1-42}$), which reduces the binding of YY1 to the *Fuzzy* gene promoter and upregulates the expression of the Fuzzy protein[8]. An increase in Fuzzy expression triggers the activation of a mitogen-activated protein kinase (MAPK)–caspase pro-apoptotic signalling cascade and results in neuronal cell death[8].

Wnt signalling controls numerous essential cellular behaviours such as cell proliferation, polarity and mobility[19]. Depending on the nuclear factors involved, Wnt signalling can be categorised into three major pathways: the canonical Wnt/β-catenin pathway and the non-canonical Wnt/PCP and Wnt/Ca$^{2+}$ pathways[20]. Several lines of evidence highlight the role of Wnt signalling in human neurodegenerative disorders. For example, in Alzheimer's disease models, activating the Wnt/β-catenin pathway ameliorates neurodegeneration by inhibiting Aβ$_{1-42}$ production and tau protein hyperphosphorylation[21]. Interestingly, tight regulation of the Wnt pathway by Fuzzy has been shown to be essential for embryonic maturation. Embryonic fibroblasts isolated from *Fuzzy*$^{-/-}$ mice demonstrated hyperactivation of the canonical Wnt/β-catenin pathway[22]. The expression of Wnt/β-catenin-targeted genes was reported to be upregulated in *Fuzzy*$^{-/-}$ mouse embryos[23]. However, the relationship between Fuzzy and Wnt signalling remains to be established in the context of human neurodegenerative disorders.

In this study, we demonstrated that *Fuzzy* gene transcription is downregulated in C9ALS/FTD-induced pluripotent stem cell (iPSC)-derived spinal motor neurons. Mechanistically, we demonstrated that mutant GGGGCC RNA, and not dipeptide repeat (DPR) proteins or *C9orf72* haploinsufficiency, leads to the downregulation of *Fuzzy* expression. We further showed that this downregulation involves two YY1-binding sites (*Fuzzy*$^{YY1-a}$ and *Fuzzy*$^{YY1-b}$) in the *Fuzzy* promoter. Mutant GGGGCC RNA interacts with YY1 and prevents it from binding to both *Fuzzy*$^{YY1-a}$ and *Fuzzy*$^{YY1-b}$ sites. The reduction in Fuzzy protein expression induces the expression of the Wnt/β-catenin-responsive gene Paired-like Homeodomain 2 (*PITX2*). We further showed that silencing *PITX2* expression rescues cell death and synaptic defects in our C9ALS/FTD models. This work provides the link between Wnt/β-catenin pathway activation and C9ALS/FTD pathogenicity. Taken together, our data revealed a C9ALS/FTD pathogenic pathway initiated by mutant GGGGCC RNA, which induces synaptic deficits and neuronal cell death. We also showed that targeting this pathway ameliorates C9ALS/FTD-associated neurodegeneration.

## Results

### *Fuzzy* expression is downregulated in C9ALS/FTD spinal motor neurons

We differentiated both control and C9ALS/FTD iPSCs into spinal motor neurons (Supplementary Fig. 1a[24]), and examined the transcript levels of CPLANE genes[13], *Fuzzy*, *Inturned*, *Wdpcp*, *Rsg1* and *Jbts17*, in four independently derived patient neurons. When compared with the healthy control iPSC-derived spinal motor neurons, *Fuzzy* is the only CPLANE gene whose expression was significantly reduced in C9ALS/FTD iPSC-derived spinal motor neurons (Fig. 1a; Supplementary Fig. 1b–e). A similar reduction was further detected in a pair of patient

neurons and their GGGGCC repeat expansion-corrected isogenic control neuron (Fig. 1a[25]). We also showed that Fuzzy protein level was reduced in C9ALS/FTD iPSC-derived spinal motor neurons (Fig. 1b). In contrast to the C9ALS/FTD samples, spinal motor neurons derived from patient iPSCs carrying *TARDBP* mutations[24] showed no alteration in *Fuzzy* expression (Supplementary Fig. 1f). These results suggest that *Fuzzy* downregulation is linked to the GGGGCC repeat expansion mutation in C9ALS/FTD spinal motor neurons.

### Expression of GGGGCC RNA leads to the downregulation of *Fuzzy* promoter activity

To delineate the mechanism underlying the downregulation of *Fuzzy* transcription in C9ALS/FTD neurons (Fig. 1a), we isolated a ~3.3-kb fragment from the human *Fuzzy* promoter sequence (*Fuzzy*$^{-2732/+574}$) and inserted it into a promoter-less luciferase reporter construct (Fig. 1c). The *pAG3-(GGGGCC)$_{2/66}$* expression construct was used to investigate the effect of GGGGCC repeat expansion on *Fuzzy* promoter activity in an SK-N-MC human neuroblastoma cell model (Fig. 1c). The expanded *pAG3-(GGGGCC)$_{66}$* construct has previously been shown to produce both GGGGCC RNA and DPR proteins[26]. When co-expressed with this construct, *Fuzzy*$^{-2732/+574}$ luciferase activity was decreased, indicating GGGGCC repeat expansion downregulates *Fuzzy* promoter activity (Fig. 1d). A similar effect was detected in luciferase constructs that harboured the *Fuzzy*$^{-2032/+574}$ and *Fuzzy*$^{-1332/+574}$ regions, but not the *Fuzzy*$^{-632/+574}$ region (Fig. 1d). This indicates that the *Fuzzy*$^{-1332/-633}$ region is crucial for the GGGGCC-mediated downregulation of *Fuzzy* transcription. We also demonstrated that *Fuzzy* promoter activity inhibition is GGGGCC repeat length-dependent (Supplementary Fig. 2a, b). As mutant GGGGCC RNA, DPR proteins and *C9orf72* haploinsufficiency are all known to contribute to C9ALS/FTD pathogenesis[27], we next assessed their respective contributions to *Fuzzy* promoter activity perturbation. We found that expression of the expanded GGGGCC RNA alone (*pcDNA3.1(+)-(GGGGCC)$_{106}$-RO*; Fig. 1c), but not of the DPR proteins (*pUltra-poly(GA/GR/PR)$_{50}$*; Supplementary Fig. 2a) or knockdown of *C9orf72* (Supplementary Fig. 2a), inhibited *Fuzzy* promoter activity (Fig. 1e; Supplementary Fig. 2c, d). Treating *(GGGGCC)$_{106}$-RO*-expressing cells with BIND, a potent GGGGCC RNA-binding peptide[26], restored *Fuzzy* promoter activity (Fig. 1e). These data clearly show that *Fuzzy* promoter activity is dysregulated by mutant GGGGCC RNA.

Upon the deletion of the *Fuzzy*$^{-1332/-1143}$ region, the mutant GGGGCC RNA's effect on *Fuzzy* promoter activity was diminished (Fig. 1f). Based on an in silico prediction, a putative transcriptional factor binding site, *Fuzzy*$^{YY1-a}$ (Supplementary Fig. 2e), was identified. We took a site-directed mutagenesis approach to examine the involvement of *Fuzzy*$^{YY1-a}$ in the mutant GGGGCC RNA-mediated downregulation of *Fuzzy* promoter activity. When a single nucleotide substitution, T to C, was introduced to the *Fuzzy*$^{YY1-a}$ site (Fig. 1g[28]), the *(GGGGCC)$_{66}$*-mediated downregulation of *Fuzzy* promoter activity was abolished (Fig. 1h). We previously identified another YY1 binding site, *Fuzzy*$^{YY1-b}$, in the *Fuzzy* promoter region (Fig. 1g[8]). In contrast to *Fuzzy*$^{YY1-a}$, when the same T to C substitution was introduced to *Fuzzy*$^{YY1-b}$ site (Fig. 1g), no restoration of *Fuzzy* promoter activity in *(GGGGCC)$_{66}$*-expressing cells was observed (Fig. 1i). Our data suggest that *Fuzzy*$^{YY1-a}$ is critical for the *Fuzzy* downregulation in mutant GGGGCC RNA-expressing cells.

### YY1 binds to and plays opposing roles at *Fuzzy*$^{YY1-a}$ and *Fuzzy*$^{YY1-b}$ sites

We carried out cell-free electrophoretic mobility shift assay (EMSA) to examine the binding affinities between YY1 protein and *Fuzzy*$^{YY1}$ DNAs. We found that recombinant YY1 protein formed protein–DNA complexes with both *Fuzzy*$^{YY1-a}$ and *Fuzzy*$^{YY1-b}$ DNA probes (Fig. 2a). The *AAV P5* DNA probe was used as a YY1-binding positive control (Supplementary Fig. 3a[29]). The binding affinity of YY1–*Fuzzy*$^{YY1-a}$ ($K_d = 242.40 \pm 21.66$ nM) was ~1.6-fold higher than that of YY1–*Fuzzy*$^{YY1-b}$

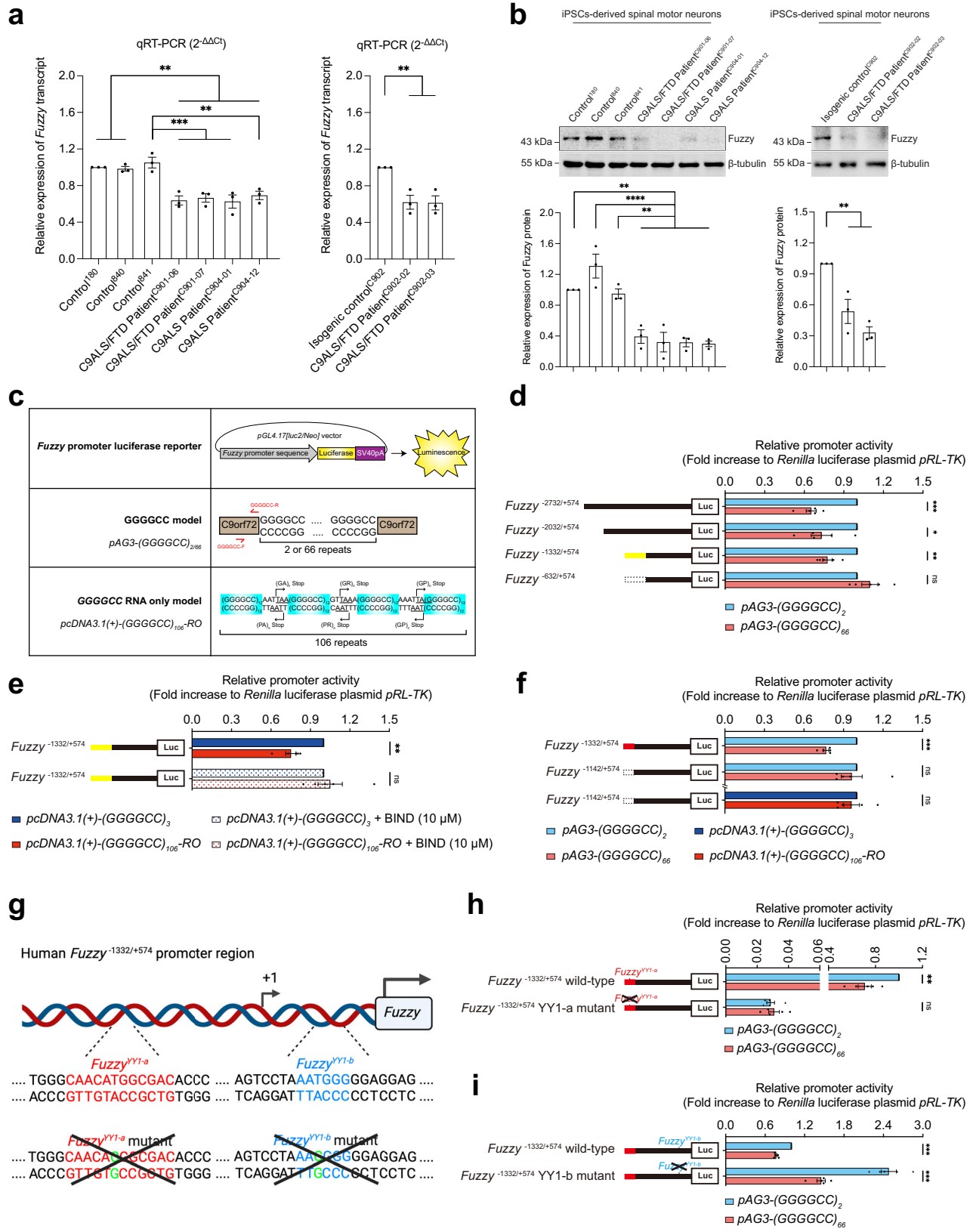

(K$_d$ = 404.90 ± 45.85 nM), indicating that YY1 binds more strongly to *Fuzzy*$^{YY1-a}$ than to *Fuzzy*$^{YY1-b}$ (Fig. 2a).

Using a cell-based chromatin immunoprecipitation (ChIP) assay, we further demonstrated the binding of YY1 to both endogenous *Fuzzy*$^{YY1-a}$ and *Fuzzy*$^{YY1-b}$ promoter sites in SK-N-MC cells (Fig. 2b;

Supplementary Fig. 3b, c). We also found that YY1 binds to the endogenous *Fuzzy* DNA sequence via its C-terminal zinc finger (ZF) domain (Fig. 2b; Supplementary Fig. 3b–d). The YY1 protein ZF domain carries four highly conserved ZF motifs (ZF1-4; Supplementary Fig. 3b, d[30,31]). We next determined which ZFs are involved in YY1's interaction with

**Fig. 1 | Expression of mutant GGGGCC RNA causes downregulation of *Fuzzy* promoter activity. a, b** The *Fuzzy*/Fuzzy transcript (**a**) and protein (**b**) levels were downregulated in C9ALS/FTD patient iPSC-derived spinal motor neurons when compared to the healthy and isogenic control neurons. **c** Illustration of the *Fuzzy* promoter luciferase reporter and different GGGGCC expression constructs used in this study. **d** Overexpression of pAG3-$(GGGGCC)_{66}$ construct suppressed *Fuzzy* promoter luciferase activities at $Fuzzy^{-2732/+574}$, $Fuzzy^{-2032/+574}$ and $Fuzzy^{-1332/+574}$ regions, whereas the activity of $Fuzzy^{-632/+574}$ luciferase construct was not altered. The $(GGGGCC)_{66}$-responsive region, $Fuzzy^{-1332/-633}$, is in yellow. **e** *Fuzzy* promoter activity was downregulated in $(GGGGCC)_{106}$-RO-expressing cells, while such down-regulation was restored upon treatment of BIND peptide. **f** The $(GGGGCC)_{66}$- and $(GGGGCC)_{106}$-RO-responsive element was further narrowed down to $Fuzzy^{-1332/-1143}$ region, which is in red. **g** Schematic representation of the locations and sequences of $Fuzzy^{YY1-a}$ (in red font) and $Fuzzy^{YY1-b}$ (in blue font) sites in $Fuzzy^{-1332/+574}$ promoter region. "+1" represents the transcriptional initiation (arrow) site. The $Fuzzy^{YY1-a}$ and $Fuzzy^{YY1-b}$ sites were mutated upon introducing a single nucleotide substitution from "T" to "C" (in green font). **h, i** Mutation of $Fuzzy^{YY1-a}$ (**h**), but not $Fuzzy^{YY1-b}$ (**i**) abolished the $(GGGGCC)_{66}$-mediated downregulation of $Fuzzy^{-1332/+574}$ promoter activity. One-way ANOVA followed by *post hoc* Tukey's test was used for the comparisons between disease and healthy control neurons in panels (**a**) and (**b**). One-way ANOVA followed by *post hoc* Dunnett's test was used for the comparisons between disease and isogenic control neurons in panels (**a**) and (**b**). Two-tailed unpaired Student's *t*-test was used in panels (**d**–**f**), (**h**) and (**i**). The exact *P* values are listed in Supplementary Table 7. For panels (**a**) and (**b**), *n* = 3 biologically independent experiments. For the rest of the panels, *n* = 5 biologically independent experiments. Data is presented as mean ± SEM. The illustrations in panels (**c**) and (**g**) were created using Adobe Illustrator and BioRender.com, respectively. Source data are provided as a Source Data file.

$Fuzzy^{YY1-a}$ and $Fuzzy^{YY1-b}$. It was found that both full-length YY1 and $YY1^{ZF1+ZF2+ZF3}$ mutant proteins showed comparable binding capacities towards $Fuzzy^{YY1-a}$. This indicates that ZF4 is not necessary for mediating YY1–$Fuzzy^{YY1-a}$ interaction. In contrast, all mutant YY1 proteins that lack ZF3, including $YY1^{ZF1+ZF2}$, $YY1^{ZF1}$ and $YY1^{\Delta ZF}$, showed reduced $Fuzzy^{YY1-a}$ binding (Fig. 2b). This shows that the ZF3 motif of YY1 is a key determinant of $Fuzzy^{YY1-a}$ binding. Using the same approach, we found that the interaction between YY1 and $Fuzzy^{YY1-b}$ is mediated through ZF4 (Fig. 2b). These results highlight the distinct roles of ZF motifs in facilitating YY1–$Fuzzy^{YY1-a}$ and YY1–$Fuzzy^{YY1-b}$ interactions.

YY1 controls gene transcription upon binding to gene promoters[32]. We studied the regulatory function of YY1 on *Fuzzy* transcription by introducing a known YY1–DNA binding disruption mutation[28] to $Fuzzy^{YY1}$ sites (Fig. 1g). Inactivation of the $Fuzzy^{YY1-a}$ site led to a dramatic ~50-fold reduction in *Fuzzy* promoter activity in SK-N-MC cells (Fig. 2c). In contrast, inactivating $Fuzzy^{YY1-b}$ site only mildly potentiated *Fuzzy* promoter activity by ~2.5 fold (Fig. 2c). Our data clearly demonstrate that YY1 interacts with the $Fuzzy^{YY1-a}$ site to promote transcription activation, while it serves as a transcriptional repressor at the $Fuzzy^{YY1-b}$ site.

### Reducing the binding of YY1 to $Fuzzy^{YY1}$ promoter sites limits *Fuzzy* promoter activity

We showed that mutant GGGGCC RNA downregulates *Fuzzy* transcription (Fig. 1a). Similarly, knockdown of *YY1* (Supplementary Fig. 3e) caused inhibition of wild-type *Fuzzy* promoter activity (Fig. 2d). We next examined whether GGGGCC RNA would disrupt YY1–$Fuzzy^{YY1}$ interaction, which consequently leads to the reduction of *Fuzzy* transcripts in patient neurons (Fig. 1a). To test this, we first performed EMSA to examine whether GGGGCC RNA perturbs the YY1–$Fuzzy^{YY1}$ binding. Upon the addition of GGGGCC RNA, the binding of YY1 to both $Fuzzy^{YY1-a}$ and $Fuzzy^{YY1-b}$ sites were disrupted (Fig. 2e). When compared with the YY1–$Fuzzy^{YY1-a}$ complex formation, it is evident that the YY1–$Fuzzy^{YY1-b}$ complexes were more susceptible to GGGGCC RNA disturbance (Fig. 2e). We next performed cell-based ChIP assay to validate our cell-free EMSA findings, and found that the binding of YY1 to both $Fuzzy^{YY1}$ sites was disrupted in the mutant $(GGGGCC)_{66}$ RNA-expressing cells (Fig. 2f; Supplementary Fig. 3f). Under the same mutant $(GGGGCC)_{66}$ expression condition, the YY1–$Fuzzy^{YY1-a}$ interaction showed a higher tolerance toward mutant GGGGCC RNA than YY1–$Fuzzy^{YY1-b}$ interaction (Fig. 2f). This is in line with our other observation (Fig. 2a) that YY1 has a higher binding affinity for $Fuzzy^{YY1-a}$ than for $Fuzzy^{YY1-b}$. In summary, our findings support the notion that when compared to $Fuzzy^{YY1-a}$, the binding of YY1 to $Fuzzy^{YY1-b}$ is more susceptible to mutant GGGGCC RNA disturbance (Fig. 2e, f).

We also examined the consequence of mutant GGGGCC RNA-mediated disruption of YY1 binding on $Fuzzy^{YY1}$ sites. We observed that a stepwise increase in mutant GGGGCC RNA expression caused a progressive disruption of YY1 binding to both $Fuzzy^{YY1-a}$ and $Fuzzy^{YY1-b}$ sites (Fig. 2f). Interestingly, such upward mutant GGGGCC RNA gradient also led to a gradual inhibition of *Fuzzy* promoter activity in our SK-N-MC cell model (Fig. 2g). In Fig. 2c, we showed a dramatic ~50-fold reduction of *Fuzzy* promoter activity when the YY1–$Fuzzy^{YY1-a}$ interaction was perturbed. On the contrary, only a mild induction (~2.5-fold) of *Fuzzy* activity was detected when the YY1–$Fuzzy^{YY1-b}$ interaction was abolished (Fig. 2c). Collectively, our data suggest that the mutant GGGGCC RNA-mediated downregulation of *Fuzzy* transcription is mainly attributable to the impairment of YY1–$Fuzzy^{YY1-a}$ binding, which translates into a pronounced inhibition of *Fuzzy* promoter activity. The mild induction effect mediated through the $Fuzzy^{YY1-b}$ site would likely be overshadowed by the $Fuzzy^{YY1-a}$ site.

### GGGGCC RNA interacts with and sequesters YY1 to RNA foci

Other than interacting with DNAs, the YY1 protein also binds RNAs[33]. We continued to investigate whether GGGGCC RNA would interact with YY1. Our EMSA analysis showed that recombinant $YY1^{full-length}$ protein bound directly to GGGGCC RNA at $K_d = 46.76 \pm 3.42$ nM (Fig. 3a), but less tightly to a randomised GC-rich RNA probe ($K_d = 200.60 \pm 17.92$ nM; Supplementary Fig. 4a) that carries equivalent number of G and C nucleotides as with our GGGGCC RNA probe (Supplementary Table 4). Similar to EMSA, the fluorescence polarisation (FP) assay has been widely used to study protein-nucleic acid interactions[34]. Consistent with EMSA, our FP assay results also demonstrate a direct binding between recombinant $YY1^{full-length}$ protein and GGGGCC RNA ($K_d = 45.30 \pm 3.67$ nM; Fig. 3b). Notably, the binding affinity of YY1–GGGGCC RNA ($K_d = 46.76 \pm 3.42$ nM) was much stronger than that of YY1–$Fuzzy^{YY1-a}$ ($K_d = 242.40 \pm 21.66$ nM) and YY1–$Fuzzy^{YY1-b}$ ($K_d = 404.90 \pm 45.85$ nM) (Fig. 2a). Our data therefore support the notion that GGGGCC RNA interacts with YY1 and prevents it from binding to $Fuzzy^{YY1}$ sites (Fig. 2e, f). This consequently leads to the downregulation of *Fuzzy* transcription (Fig. 2g). The *PP7* RNA was used as a YY1-binding positive control (Supplementary Fig. 4b[33]). We also detected a lower binding affinity between $YY1^{full-length}$ protein and GGGGCC DNA (Supplementary Fig. 4c).

The expression of mutant GGGGCC RNA leads to the formation of RNA foci in SK-N-MC cells (Fig. 3c[35]). Using in situ hybridisation followed by immunofluorescence, we showed that GGGGCC RNA foci co-localised with endogenous heterogeneous nuclear ribonucleoprotein H1 (hnRNP H), a known GGGGCC RNA-interacting protein (Fig. 3c; Supplementary Fig. 5a[36]). Such a pattern of co-localisation was detected in ~80% of GGGGCC RNA foci-positive cells (Fig. 3f). Under the same experimental conditions, the co-localisation between endogenous YY1 and GGGGCC RNA foci was detected in ~40% of GGGGCC RNA foci-positive cells (Fig. 3d, f; Supplementary Fig. 5b). The subcellular co-localisation of YY1 protein with GGGGCC RNA foci was further confirmed using a separate anti-YY1 antibody that recognises a different epitope (Fig. 3e, f; Supplementary Fig. 5c). It is of note that the protein-foci co-localisation pattern was abolished upon treatment with RNase (Fig. 3c–e). This indicates that YY1 protein mis-localisation is RNA-mediated.

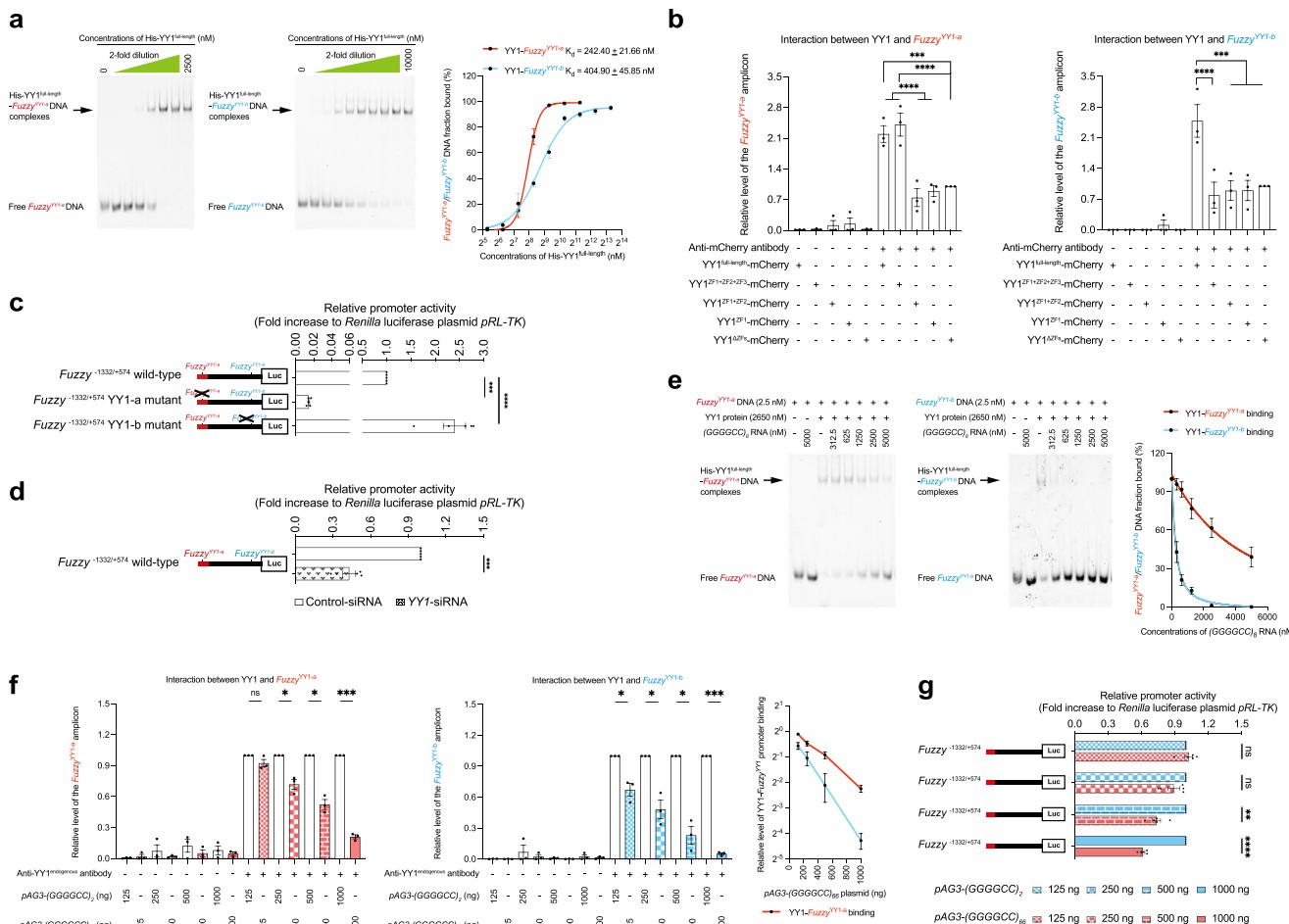

**Fig. 2 | GGGGCC RNA perturbs the binding of YY1 to *Fuzzy* promoter, leading to *Fuzzy* downregulation. a** The binding of recombinant YY1 protein to *Fuzzy*[YY1-a] DNA is stronger than to *Fuzzy*[YY1-b] DNA. The $R^2$ of the YY1–*Fuzzy*[YY1-a] and YY1–*Fuzzy*[YY1-b] binding curves are 0.9915 and 0.9818, respectively. The *Fuzzy* promoter region that carries one copy of *Fuzzy*[YY1-a] or *Fuzzy*[YY1-b] site was synthesized and used in the EMSA. The sequences of the Cy5-labelled *Fuzzy*[YY1-a] DNA and *Fuzzy*[YY1-b] DNA (each of 20 base pairs in length) probes are listed in Supplementary Table 4. **b** The ZF3 in YY1 is required for interacting with *Fuzzy*[YY1-a], while ZF4 mediates the interaction between YY1 and *Fuzzy*[YY1-b]. Both *Fuzzy*[YY1-a] and *Fuzzy*[YY1-b] DNAs were amplified from the endogenous *Fuzzy* gene promoter. **c** Mutation of *Fuzzy*[YY1-a] led to downregulation of *Fuzzy* promoter activity, whereas mutation of *Fuzzy*[YY1-b] caused the opposite upregulation effect. **d** Knockdown of *YY1* led to the inhibition of *Fuzzy* promoter activity. **e** Addition of GGGGCC RNA perturbed the formation of protein–DNA complexes between recombinant YY1 protein and *Fuzzy*[YY1-a]/*Fuzzy*[YY1-b] DNA, leading to the

increase in levels of free *Fuzzy*[YY1-a] and *Fuzzy*[YY1-b] DNAs. The YY1–*Fuzzy*[YY1-b] binding is more susceptible to GGGGCC RNA interference. **f** The binding of endogenous YY1 protein to *Fuzzy*[YY1-a] and *Fuzzy*[YY1-b] sites was impaired upon overexpression of the *pAG3-(GGGGCC)66* construct. The YY1–*Fuzzy*[YY1-b] interaction was more prone to be disturbed in the presence of mutant GGGGCC RNA. Both *Fuzzy*[YY1-a] and *Fuzzy*[YY1-b] DNAs were amplified from the endogenous *Fuzzy* promoter. **g** The increase of the transfection amount of *pAG3-(GGGGCC)66* plasmid gradually inhibited *Fuzzy* promoter activity. One-way ANOVA followed by *post hoc* Tukey's test and one-way ANOVA followed by *post hoc* Dunnett's test were used in panels (**b**) and (**c**), respectively. Two-tailed unpaired Student's *t*-test was used in panels (**d**), (**f**) and (**g**). The exact *P* values are listed in Supplementary Table 7. For panels (**c**), (**d**) and (**g**), *n* = 5 biologically independent experiments. For the rest of the panels, *n* = 3 biologically independent experiments. Data is presented as mean ± SEM. Source data are provided as a Source Data file.

We further tested whether the co-localisation between YY1 and GGGGCC RNA foci decreased the mobility of YY1. Twelve copies of MS2 aptamers, which serve as binding sites for MS2-coat binding protein (MS2CP)[37], were inserted downstream of the *pAG3-(GGGGCC)2/66* construct (Fig. 1c). Live cell imaging was performed on cells co-transfected with *pAG3-(GGGGCC)66-MS2* and *MS2CP-YFP* constructs. The GGGGCC RNA foci were found to be co-localised with the fluorescently tagged YY1 protein (YY1-mCherry; Fig. 3g). We next performed fluorescence recovery after photobleaching assay on these cells and found a lack of fluorescent recovery of the YY1-mCherry protein upon photobleaching in GGGGCC RNA foci-positive cells (Fig. 3g). Similar findings were observed for another GGGGCC foci-recruited protein hnRNP H (Fig. 3h[36]). As a control, we demonstrated that the co-transfection of *pAG3-(GGGGCC)2-MS2* unexpanded control and *MS2CP-YFP* did not result in any detectable RNA foci formation (Fig. 3g). These results show that the intracellular mobility

of YY1 was compromised in mutant GGGGCC RNA cells, presumably due to its association with the GGGGCC RNA and/or sequestration to the RNA foci.

## YY1[ZF3] is involved in mediating YY1–GGGGCC RNA interaction

Zinc finger motifs of YY1 have been reported to mediate protein-RNA interactions[33]. We performed cell-free EMSA using two versions of recombinant mutant YY1 proteins, one carries ZF1-3 motifs (YY1[ZF1+ZF2+ZF3]) and another one with only ZF1 and 2 (YY1[ZF1+ZF2]) (Supplementary Fig. 6a), to determine the role of ZF motifs in YY1–GGGGCC RNA binding. When compared with YY1[full-length] protein ($K_d = 46.76 \pm 3.42$ nM; Fig. 3a), YY1[ZF1+ZF2+ZF3] interacted with (GGGGCC)8 RNA at a similar binding affinity ($K_d = 59.01 \pm 1.23$ nM; Fig. 4a). This suggests that the ZF4 motif is not crucial for GGGGCC RNA interaction. In contrast, the binding affinity between YY1[ZF1+ZF2] and (GGGGCC)8 RNA was reduced to $129.43 \pm 26.23$ nM (Fig. 4a). Our data thus highlight that

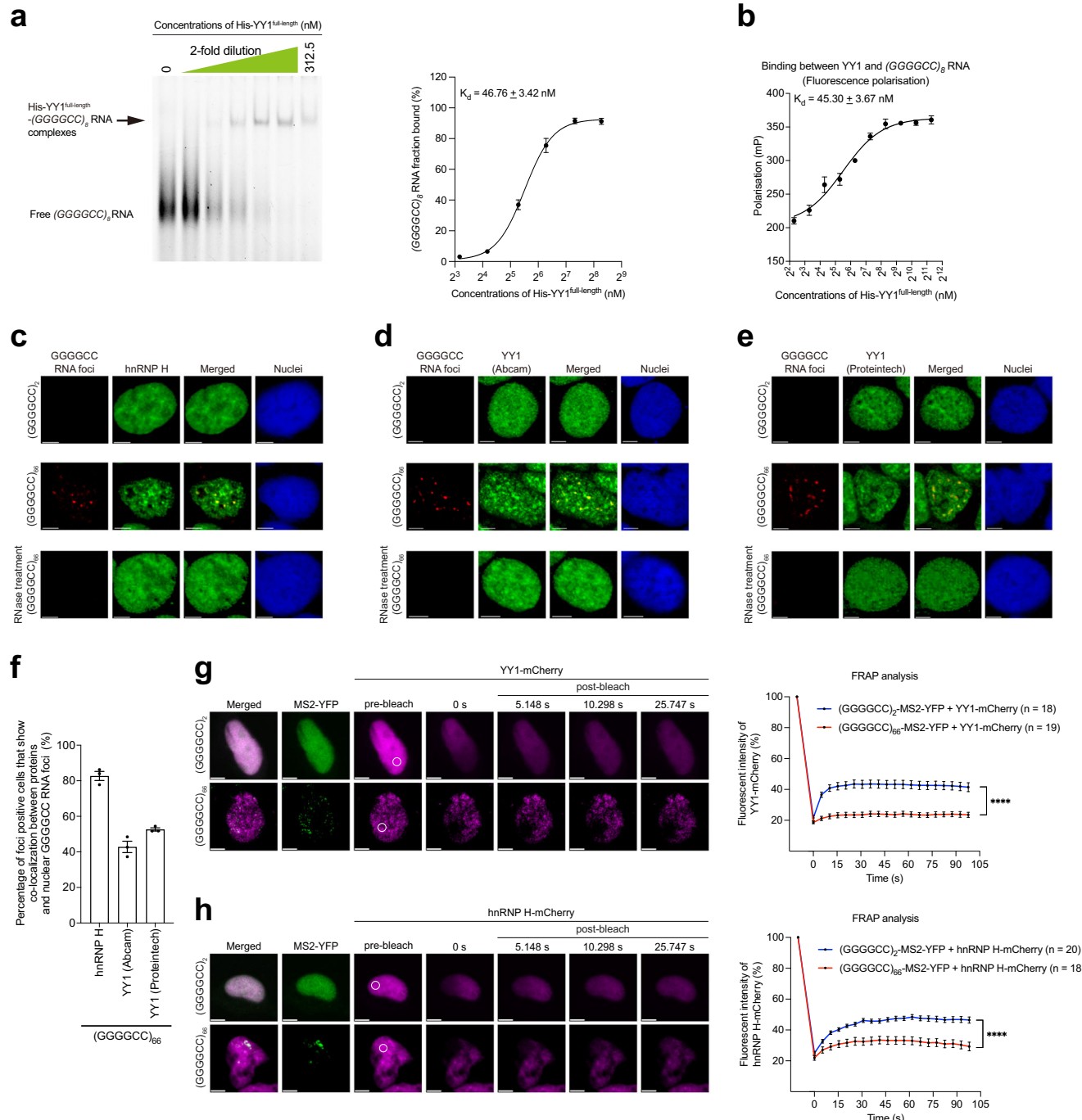

**Fig. 3 | The GGGGCC RNA binds to YY1 protein and recruits it to RNA foci. a** The binding between recombinant His-YY1[full-length] protein and $(GGGGCC)_8$ RNA was detected by EMSA. The $R^2$ of the YY1−$(GGGGCC)_8$ RNA binding curve is 0.9914. The sequence of the Cy5-labelled $(GGGGCC)_8$ RNA (48 bases in length) probe is listed in Supplementary Table 4. **b** The fluorescence polarisation assay was performed to demonstrate the binding between recombinant His-YY1[full-length] protein and $(GGGGCC)_8$ RNA. The $R^2$ of the YY1−$(GGGGCC)_8$ RNA binding curve is 0.9881. The sequence of the FAM-labelled $(GGGGCC)_8$ RNA (50 bases in length) probe is listed in Supplementary Table 4. **c** The endogenous hnRNP H protein (green) was found co-localised with GGGGCC RNA foci (red) in $(GGGGCC)_{66}$-expressing SK-N-MC cells. Scale bars: 5 µm. No GGGGCC RNA foci were detected in $(GGGGCC)_2$-expressing cells or cells that received RNase treatment. **d, e** The co-localisation between endogenous YY1 protein (green) and GGGGCC RNA foci (red) was detected in $(GGGGCC)_{66}$-transfected SK-N-MC cells by two anti-YY1 antibodies raised against different epitopes. Scale bars: 5 µm. No GGGGCC RNA foci were detected in

$(GGGGCC)_2$-transfected cells or cells that received RNase treatment. **f** is the quantification of (**c**–**e**). The number of GGGGCC foci-positive cells counted over three independent experiments were: hnRNP H: 180; YY1 (Abcam): 299; YY1 (Proteintech): 182. **g, h** The formation of GGGGCC RNA foci was detected when *pAG3-(GGGGCC)_{66}-MS2*, but not pAG3-*(GGGGCC)_2-MS2*, was co-transfected with *MS2CP-YFP* in SK-N-MC cells. The recovery of YY1-mCherry (**g**) and hnRNP H-mCherry (**h**) fluorescent signals after photobleaching was much less effective in GGGGCC foci formation cells. The circled regions indicate the region of interest (ROI) that underwent photobleaching. The pre-bleach and post-bleach fluorescent signals in the ROI were recorded. n represents the total number of cells examined over three independent experiments. Two-tailed unpaired Mann−Whitney *U* test was used in panels (**g**) and (**h**). The exact *P* values are listed in Supplementary Table 7. *n* = 3 biologically independent experiments and data is presented as mean ± SEM. Source data are provided as a Source Data file.

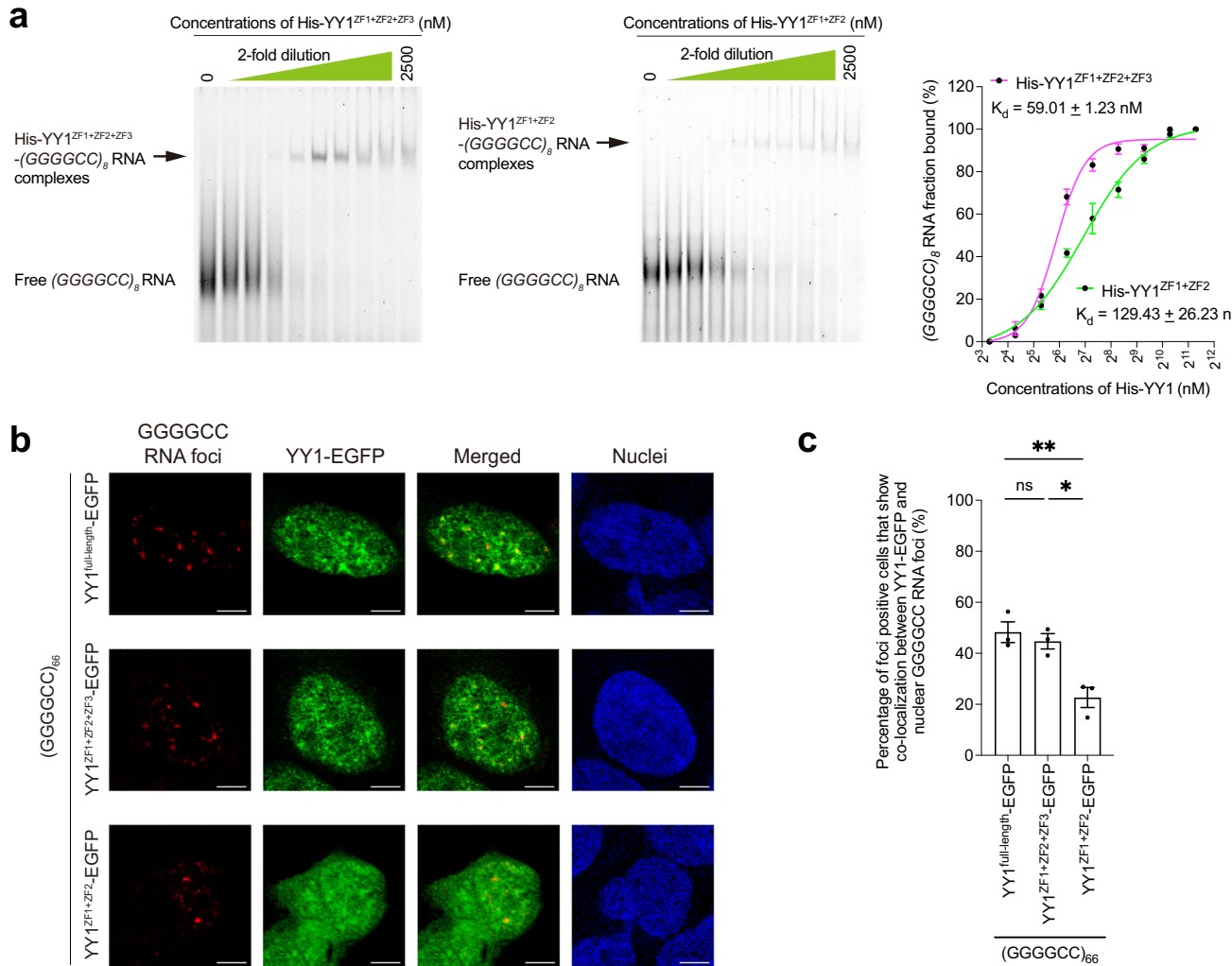

**Fig. 4 | The YY1$^{ZF3}$ plays an important role in YY1–GGGGCC RNA binding. a** The binding affinity of YY1$^{ZF1+ZF2+ZF3}$–$(GGGGCC)_8$ RNA ($K_d = 59.01 \pm 1.23$ nM) was comparable to that of YY1$^{full-length}$–$(GGGGCC)_8$ RNA ($K_d = 46.76 \pm 3.42$ nM; Fig. 3a) and was stronger than that of YY1$^{ZF1+ZF2}$–$(GGGGCC)_8$ RNA ($K_d = 129.43 \pm 26.23$ nM). The R$^2$ of the YY1$^{ZF1+ZF2+ZF3}$–$(GGGGCC)_8$ RNA and YY1$^{ZF1+ZF2}$–$(GGGGCC)_8$ RNA binding curves are 0.9871 and 0.9828, respectively. The sequence of the Cy5-labelled $(GGGGCC)_8$ RNA (48 bases in length) probe is listed in Supplementary Table 4. **b** The co-localisation between YY1 (green) and GGGGCC RNA foci (red) was diminished when the YY1$^{ZF3}$,

but not YY1$^{ZF4}$, was deleted. Scale bars: 5 μm. **c** is the quantification of (**b**). The number of GGGGCC foci-positive and YY1-EGFP co-transfected cells counted over three independent experiments were: YY1$^{full-length}$-EGFP: 217; YY1$^{ZF1+ZF2+ZF3}$-EGFP: 187; YY1$^{ZF1+ZF2}$-EGFP: 106. One-way ANOVA followed by *post hoc* Tukey's test was used in (**c**). The exact *P* values are listed in Supplementary Table 7. *n* = 3 biologically independent experiments and data is presented as mean ± SEM. Source data are provided as a Source Data file.

the YY1$^{ZF3}$ motif is important for mediating YY1 and GGGGCC RNA interaction. We further found that the ZF3 motif is one determinant for YY1 to co-localise with RNA foci (Fig. 4b, c; Supplementary Fig. 6b).

### YY1 overexpression ameliorates cell death and synaptic defects in C9ALS/FTD models

The GGGGCC RNA foci were detected in C9ALS/FTD iPSC-derived spinal motor neurons, but not in isogenic or healthy control neurons (Fig. 5a; Supplementary Fig. 7, 8). We showed that these RNA foci co-localised with endogenous YY1 (Fig. 5a, b; Supplementary Fig. 7, 8) and hnRNP H (Supplementary Fig. 9) proteins. Such protein-RNA co-localisation pattern was abolished upon RNase treatment (Fig. 5a; Supplementary Figs. 7–9). When compared with isogenic control neurons, the binding of endogenous YY1 protein to endogenous *Fuzzy*$^{YY1}$ promoter regions was impaired in C9ALS/FTD neurons. YY1-mCherry overexpression in disease neurons ameliorated the defects in YY1–*Fuzzy*$^{YY1}$ binding (Fig. 6a; Supplementary Fig. 10a). Under an overexpression condition, isogenic control neurons exogenously supplemented with YY1 also showed an enhanced YY1–*Fuzzy*$^{YY1}$ binding (Fig. 6a). We next performed a nascent

RNA capture assay[38] to determine the level of nascent *Fuzzy* transcript in iPSC-derived spinal motor neuron samples. When compared with the isogenic control neurons, the level of nascent *Fuzzy* mRNA was reduced in C9ALS/FTD neurons (Fig. 6b). Intriguingly, such reduction was rescued upon YY1-mCherry overexpression (Fig. 6b). Taken together, our data suggest that YY1 overexpression suppresses *Fuzzy* transcriptional dysregulation in C9ALS/FTD neurons.

When transfected with a construct carrying mutant GGGGCC (*pAG3-(GGGGCC)$_{66}$*), cell death became evident in our SK-N-MC cell model, as evidenced by the release of lactate dehydrogenase[26]. We found that YY1 overexpression mitigated *(GGGGCC)$_{66}$*-induced cell death (Fig. 6c). To confirm that the YY1-mediated suppression of *(GGGGCC)$_{66}$*-induced cell death was not due to a reduction of mutant GGGGCC RNA expression in the co-transfected cells, we examined mutant GGGGCC expression by RT-PCR[26] and found that GGGGCC mRNA levels were not affected upon YY1 overexpression (Supplementary Fig. 10b, c). We next used a GGGGCC *Drosophila* model to test the modulatory function of YY1 in vivo. When expressed pan-neuronally, the expanded *(GGGGCC)$_{36}$* transgene induced climbing deficits

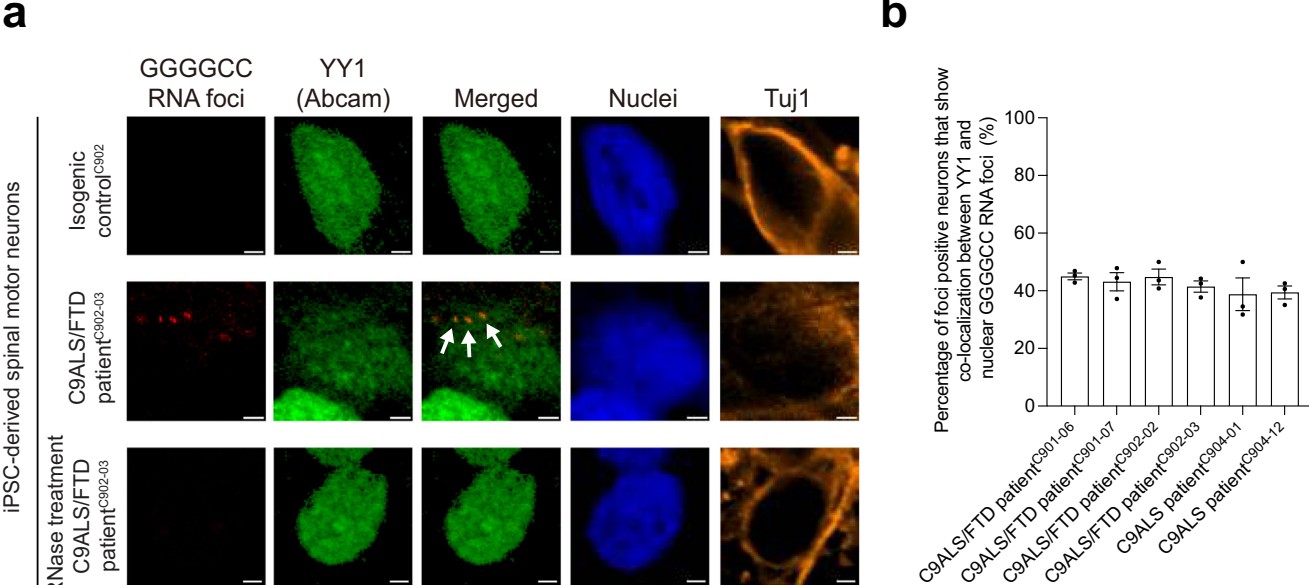

**Fig. 5 | The YY1 protein is recruited to GGGGCC RNA foci in C9ALS/FTD iPSC-derived spinal motor neurons. a** The endogenous YY1 protein (green) was recruited to the GGGGCC RNA foci (red) in C9ALS/FTD iPSC-derived spinal motor neurons. No such recruitment was detected in isogenic control spinal motor neurons or neurons that received RNase treatment. **b** is the quantification of Fig. 5a and Supplementary Fig. 8. The number of GGGGCC foci-positive neurons counted over three independent experiments were: C901-06: 127; C901-07: 103; C902-02: 122; C902-03: 152; C904-01: 176; C904-12: 143. Data are presented as mean ± SEM. No GGGGCC foci formation was detected in isogenic control iPSC-derived spinal motor neurons or neurons treated with RNase. Arrows indicate the co-localisation between the endogenous YY1 protein and GGGGCC RNA foci. The cell nuclei were stained with Hoechst 33342 (blue). Scale bars: 2 μm. Source data are provided as a Source Data file.

and shortened the lifespans of the flies[39]. Overexpression of Pho (Supplementary Fig. 10d), the *Drosophila* orthologue of human YY1[40], ameliorated the climbing and survival defects in (GGGGCC)$_{36}$ flies (Fig. 6d, e). The above findings confirm the modulatory role of YY1 as a suppressor in C9ALS/FTD-associated neurotoxicity.

Synaptic dysfunction and degeneration have been documented as typical pathogenic features of C9ALS/FTD neurons[41,42]. Bassoon and Homer1 are key pre- and postsynaptic structural proteins, respectively, that support normal synaptic function[43]. We found that both the Bassoon and Homer1 puncta numbers along the neurites were significantly decreased in C9ALS/FTD spinal motor neurons relative to isogenic control neurons (Fig. 6f–i; Supplementary Fig. 10e, f). Our finding suggests that synaptic activity in both the pre- and post-synaptic compartments of disease neurons is potentially compromised. We found that YY1 overexpression in C9ALS/FTD iPSC-derived spinal motor neurons rescued the Bassoon and Homer1 puncta numbers, suggesting a restoration of synaptic defects in C9ALS/FTD neurons (Fig. 6f–i; Supplementary Fig. 10e, f).

### Activation of the Wnt/β-catenin pathway contributes to cell death and synaptic defects in C9ALS/FTD models

Fuzzy negatively regulates canonical Wnt/β-catenin activity[22,23]. We utilised a luciferase-based reporter system (TOP/FOP Flash[44]) to examine the activity of the Wnt/β-catenin pathway in mutant GGGGCC RNA-expressing cells. Intriguingly, we found that compared with the unexpanded (GGGGCC)$_2$-transfected cells, cells transfected with the *pAG3-(GGGGCC)$_{66}$* construct showed stronger TOPFlash luciferase activity. This observation indicates the stimulation of the Wnt/β-catenin pathway in mutant GGGGCC RNA cells (Fig. 7a). Moreover, the mRNA levels of the Wnt/β-catenin targets Cyclin D1 (*CCND1*), FOS Like 1 (*FOSL1*) and Paired-like Homeodomain 2 (*PITX2*) were significantly upregulated in C9ALS/FTD neurons compared with healthy (Fig. 7b) and isogenic (Fig. 7c) control neurons. Similar changes were also detected when we analysed the transcriptomic profiles of additional C9ALS/FTD patient iPSC-derived spinal motor neuron samples from public datasets (Table 1). Consistent with transcriptomic changes, we also found that the CCND1 (Supplementary Fig. 11a), FOSL1 (Supplementary Fig. 11b) and PITX2 (Fig. 7d) protein levels were upregulated in our patient neuron model. Taken together, these data highlight the activation of the Wnt/β-catenin pathway in C9ALS/FTD neurons.

We continued to investigate the effects of CCND1, FOSL1 and PITX2 on GGGGCC-induced cell death. Of the three proteins, we found that only the knockdown of *PITX2* expression attenuated (GGGGCC)$_{66}$-induced cell death (Fig. 7e; Supplementary Fig. 11c–g). In contrast, PITX2 overexpression enhanced (GGGGCC)$_{66}$-mediated cell death in SK-N-MC cells (Supplementary Fig. 11h). These results confirm the modulatory role of PITX2 in C9ALS/FTD-associated cell death. We found that knocking down *Ptx1* (Supplementary Fig. 12a), the *Drosophila* orthologue of human *PITX2*[45], in vivo rescued the climbing and survival defects in disease flies (Fig. 7f, g). Conversely, the disease phenotypes were intensified upon *Ptx1* overexpression (Supplementary Fig. 12b–d). Furthermore, when *PITX2* expression was knocked down, the reduction in Bassoon and Homer1 puncta numbers was ameliorated in C9ALS/FTD spinal motor neurons (Fig. 7h–k; Supplementary Fig. 12e, f). In contrast, PITX2 overexpression enhanced these synaptic deficits in C9ALS/FTD spinal motor neurons (Supplementary Fig. 13a–f). In summary, our findings illustrate the role of Wnt/β-catenin pathway activation in C9ALS/FTD pathogenesis, particularly demonstrating that the induction of PITX2 expression induces cell death and synaptic defects in C9ALS/FTD neurons.

### Rescuing the dysregulation of YY1–Fuzzy–PITX2 signalling relieves synaptic defects in C9ALS/FTD neurons

Next, we investigated if the restoration of YY1–Fuzzy–PITX2 signalling would ameliorate synaptic deficits in C9ALS/FTD iPSC-derived spinal motor neurons. Overexpression of BIND, but not its scrambled version BIND-S, recovered the YY1–*Fuzzy*$^{YY1}$ promoter interactions (Fig. 8a; Supplementary Fig. 14a), followed by the restoration of Fuzzy protein

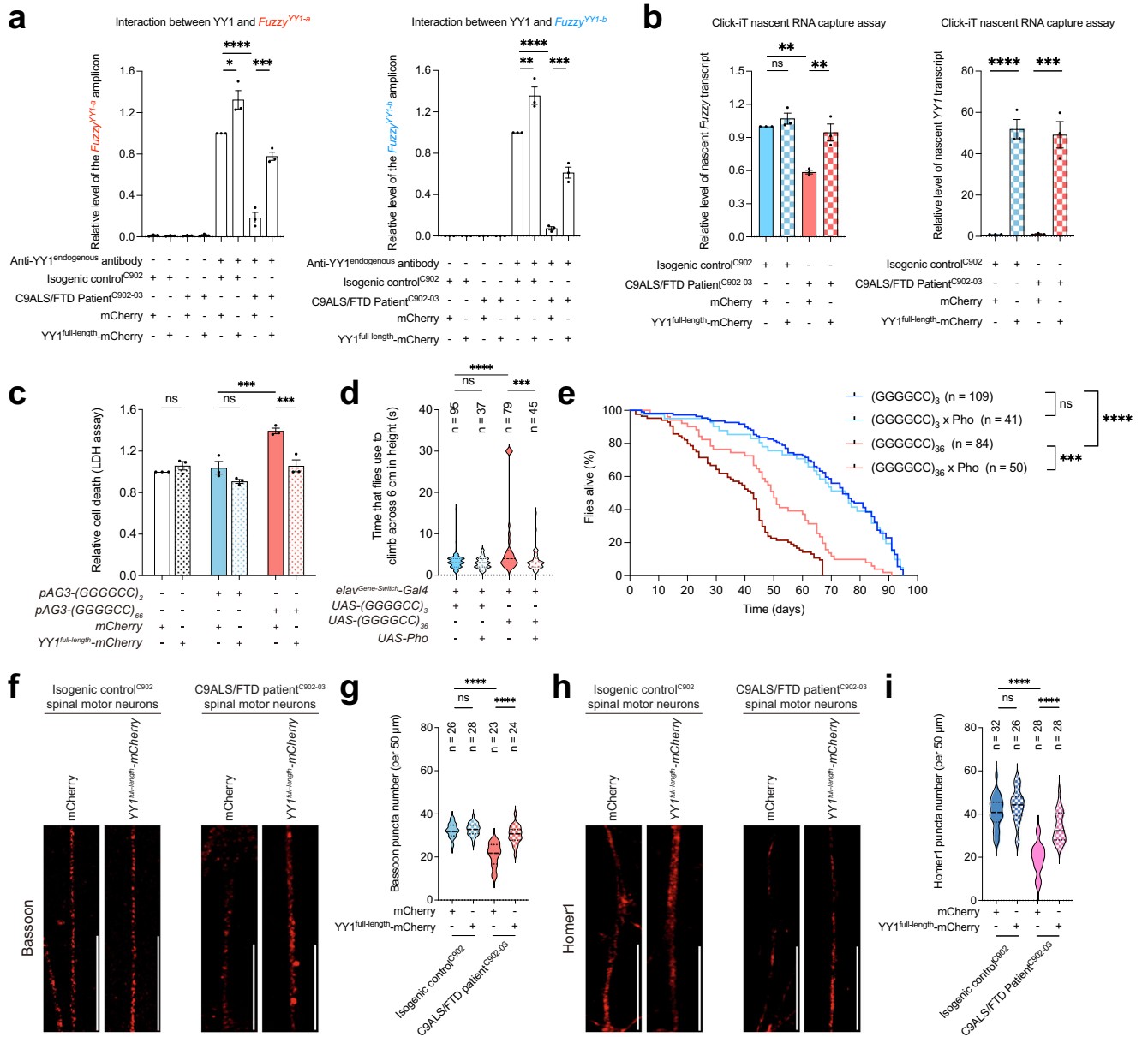

**Fig. 6 | YY1 ameliorates the cell death and synaptic defects in C9ALS/FTD models. a** The binding between YY1 and *Fuzzy*[YY1] was disrupted in C9ALS/FTD iPSC-derived spinal motor neurons, and such disruption was partially restored when YY1 was overexpressed. Overexpression of YY1 also caused the increase in YY1 and *Fuzzy*[YY1] binding in isogenic control iPSC-derived spinal motor neurons. Both *Fuzzy*[YY1-a] and *Fuzzy*[YY1-b] DNA fragments were amplified from the endogenous *Fuzzy* promoter. **b** The level of nascent *Fuzzy* transcript was reduced in C9ALS/FTD iPSC-derived spinal motor neurons relative to isogenic control neurons. Such reduction was mitigated upon overexpression of YY1. Overexpression of YY1 did not affect nascent *Fuzzy* transcript level in isogenic control neurons. YY1 overexpression caused the increase of nascent *YY1* transcript in both isogenic control and disease neurons. **c** Overexpression of YY1 suppressed the *(GGGGCC)*[66]-induced cell death. Meanwhile, it did not cause a dominant toxic effect in the untransfected or *(GGGGCC)*[2]-expressing cells. **d**, **e** The climbing (**d**) and survival (**e**) deficits of

*(GGGGCC)*[36] flies were alleviated when *Pho* was overexpressed. Overexpression of *Pho* did not affect the climbing ability and survival probability of *(GGGGCC)*[3] flies. *n* represents the total number of flies examined over three independent experiments. The genotypes of flies are listed in Supplementary Table 6. **f**–**i** Bassoon (**f**) and Homer1 (**h**) puncta numbers were reduced in C9ALS/FTD iPSC-derived spinal motor neurons. Such reduction was alleviated upon overexpression of YY1. The number of Bassoon (**f**) and Homer1 (**h**) puncta was not affected in isogenic control iPSC-derived spinal motor neurons when YY1 was overexpressed. Scale bars: 20 μm. **g** and **i** are the quantifications of **f** and **h**, respectively. *n* represents the total number of neurites examined over three independent experiments. One-way ANOVA followed by *post hoc* Tukey's test was used in panels (**a**–**d**), (**g**) and (**i**). Log-rank (Mantel-Cox) test was used in (**e**). The exact *P* values are listed in Supplementary Table 7. *n* = 3 biologically independent experiments and data is presented as mean ± SEM. Source data are provided as a Source Data file.

expression in disease neurons (Fig. 8b). PITX2 induction was suppressed upon BIND overexpression (Fig. 8c), which also rescued the decrease in Bassoon and Homer1 puncta numbers in C9ALS/FTD spinal motor neurons (Fig. 8d–g; Supplementary Fig. 14b, c). In summary, these results suggest that restoring YY1–Fuzzy–PITX2 signalling can alleviate synaptic defects in C9ALS/FTD spinal motor neurons.

## Discussion

Repeat expansion RNAs confer neurotoxicity by interacting with a range of RNA-binding proteins (RBPs), leading to the dysfunction of essential cellular pathways and, consequently, neuronal death[46]. Multiple cellular dysfunctions, such as nucleolar stress[47], nucleocytoplasmic transport defects[48] and alternative splicing deficits[49], have been reported to be

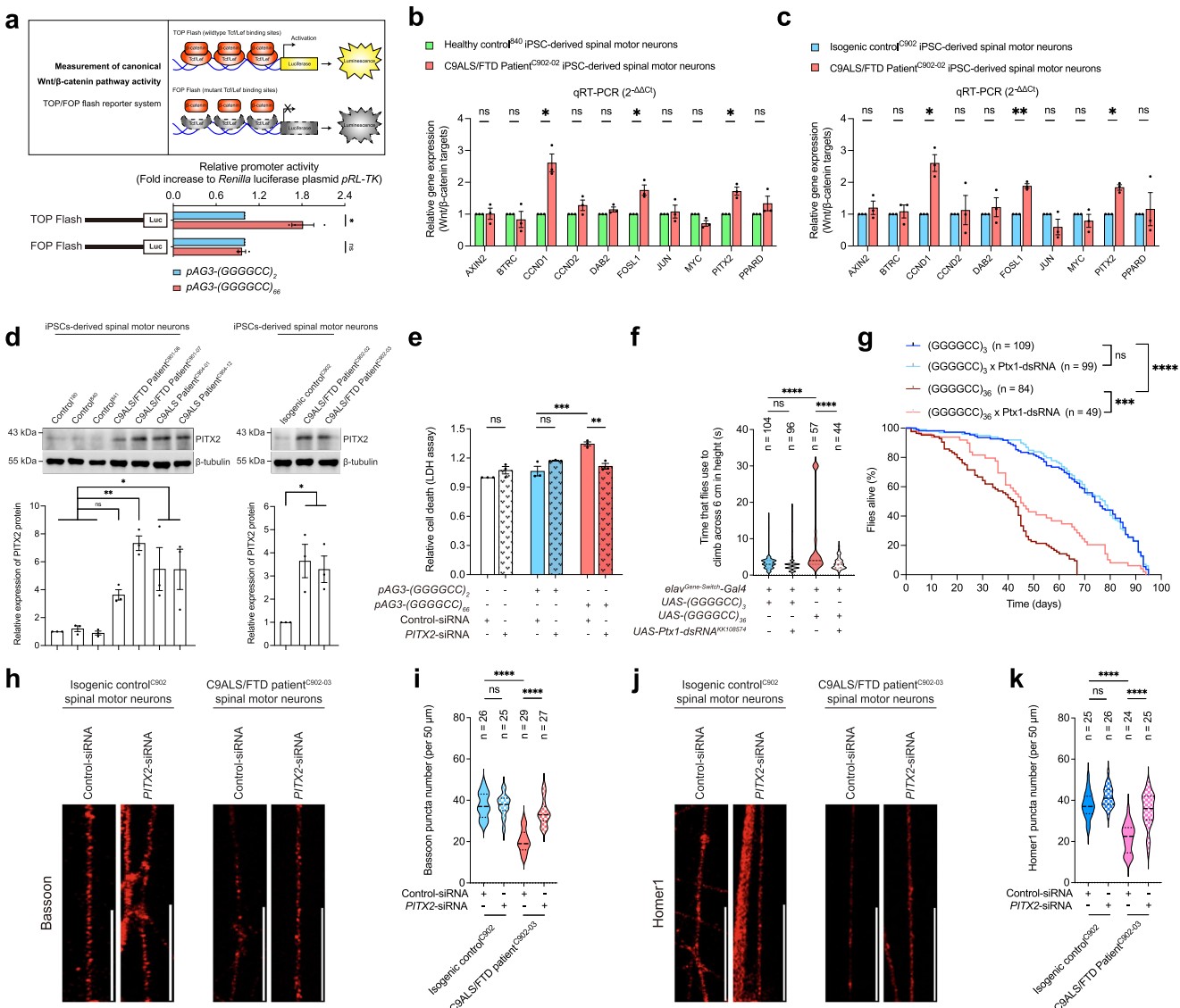

**Fig. 7 | Induction of Wnt/β-catenin target gene *PITX2* contributes to cell death and synaptic defects in C9ALS/FTD models. a** When compared to the unexpanded *(GGGGCC)₂*-expressing cells, overexpression of *pAG3-(GGGGCC)₆₆* in SK-N-MC cells stimulated the TOP flash, but not FOP flash luciferase activity. **b**, **c** The transcript levels of *CCND1*, *FOSL1*, and *PITX2*, but not other Wnt-responsive genes, were significantly upregulated in C9ALS/FTD patient iPSC-derived spinal motor neurons when compared to the healthy (**b**) and isogenic (**c**) control neurons. **d** The protein level of PITX2 was upregulated in C9ALS/FTD iPSC-derived spinal motor neurons compared to the healthy and isogenic control neurons. **e** The *(GGGGCC)₆₆*-induced cell death was suppressed upon knockdown of *PITX2*. No dominant cytotoxic effect was detected in the untransfected or *(GGGGCC)₂*-expressing cells when *PITX2* was knocked down. **f**, **g** Knockdown of *Ptx1* ameliorated the climbing (**f**) and survival (**g**) deficits of (GGGGCC)₃₆ flies, and it did not affect the climbing ability and survival probability of (GGGGCC)₃ flies. *n* represents the total number of flies examined over three independent experiments. The genotypes of flies are listed in

Supplementary Table 6. **h–k** Knockdown of *PITX2* rescued the reduction of Bassoon (**h**) and Homer1 (**j**) puncta numbers in C9ALS/FTD iPSC-derived spinal motor neurons. The number of Bassoon (**h**) and Homer1 (**j**) puncta was not affected in isogenic control iPSC-derived spinal motor neurons when *PITX2* was knocked down. Scale bars: 20 μm. **i** and **k** are the quantifications of **h** and **j**, respectively. *n* represents the total number of neurites examined over three independent experiments. Two-tailed unpaired Student's *t*-test was used in panels (**a–c**). One-way ANOVA followed by *post hoc* Tukey's test was used in panels (**d–f**), (**i**) and (**k**). One-way ANOVA followed by *post hoc* Dunnett's test was used for the comparison between disease and isogenic control neurons in (**d**). Log-rank (Mantel-Cox) test was used in panel (**g**). The exact *P* values are listed in Supplementary Table 7. *n* = 3 biologically independent experiments and data is presented as mean ± SEM. The illustration in panel 7a was created using Adobe Illustrator. Source data are provided as a Source Data file.

conferred by mutant GGGGCC RNA expression. In this study, we found that GGGGCC RNA interacts with the transcriptional regulator YY1 (Fig. 3a). Our data showed that such RNA–protein interaction impairs the binding of YY1 to the *Fuzzy* promoter (Fig. 2f), downregulates *Fuzzy* transcription (Fig. 2g) and activates the canonical Wnt/β-catenin pathway (Fig. 7a). The induction of the Wnt/β-catenin pathway target gene *PITX2* triggers cell death and synaptic defects in C9ALS/FTD models (Fig. 7). Collectively, our study uncovered the interplay among GGGGCC RNA, YY1 and *Fuzzy* promoter DNA in C9ALS/FTD neurons. We showed

that GGGGCC RNA prevents YY1 from binding to the *Fuzzy* promoter, thereby inhibiting *Fuzzy* transcription, which consequently stimulates the canonical Wnt/β-catenin pathway and leads to C9ALS/FTD-associated neurodegeneration (Fig. 9).

Emerging evidence emphasises the pathogenic roles of developmental pathways, such as the Notch, Hedgehog and Wnt/β-catenin pathways, in human neurological disorders[50]. The Wnt/β-catenin pathway has previously been studied in the context of ALS[51]. In post-mortem spinal cord samples obtained from patients with sporadic

**Table 1 | Summary of the analysis of *CCND1*, *FOSL1* and *PITX2* levels from public transcriptomic datasets**

| Reference | | C9 iPSC *n* | Control iPSC *n* |
|---|---|---|---|
| Abo-Rady et al. 2020 (PMID: 32084385) | | 1 | 1 (isogenic control) |
| **Gene name** | **P value** | **FDR value** | **Log₂ fold-change (C9 vs control)** |
| *CCND1* | 2.68E−18 | 1.31E−16 | 1.90 |
| *FOSL1* | 1.15E−06 | 1.10E−05 | 1.67 |
| **Reference** | | **C9 iPSC *n*** | **Control iPSC *n*** |
| Catanese et al. 2021 (PMID: 34125498) | | 2 | 2 (including 1 isogenic control) |
| **Gene name** | **P value** | **FDR value** | **Log₂ fold-change (C9 vs control)** |
| *CCND1* | 1.89E−08 | 5.46E−06 | 0.56 |
| *FOSL1* | 3.29E−04 | 8.12E−03 | 1.81 |
| *PITX2* | 9.04E−05 | 3.44E−03 | 4.77 |
| **Reference** | | **C9 iPSC *n*** | **Control iPSC *n*** |
| NeuroLINCS Consortium, 2021 (PMID: 34746695) | | 4 | 3 (healthy control) |
| **Gene name** | **P value** | **FDR value** | **Log₂ fold-change (C9 vs control)** |
| *CCND1* | 1.13E−04 | 8.33E−03 | 1.69 |
| *FOSL1* | 1.08E−04 | 8.10E−03 | 2.00 |
| **Reference** | | **C9 iPSC *n*** | **Control iPSC *n*** |
| NeuroLINCS Consortium, 2021 (Replication cohort; PMID: 34746695) | | 6 | 7 (healthy control) |
| **Gene name** | **P value** | **FDR value** | **Log₂ fold-change (C9 vs control)** |
| *PITX2* | 1.42E−05 | 2.78E−02 | 4.32 |

ALS, the expression of several Wnt ligands and receptors was found to be elevated[52]. Similar findings were reported in *SOD1*[G93A] ALS transgenic mouse models[53], the spinal cord samples of which showed extensive nuclear accumulation of β-catenin and induction of the target gene *CCND1*[54]. In addition to ALS, Wnt signalling has been implicated in FTD pathogenesis. Using a progranulin (*GRN*)-silenced human FTD neural stem cell model, Rosen et al.[55] reported that the transcription of Wnt signalling components was broadly dysregulated in *GRN*-inactivated cells. Similar findings have been reported by another group using FTD patient-derived iPSCs carrying a *GRN* haploinsufficiency mutation[56]. They also demonstrated that genetic correction of the mutation or treatment with a Wnt signalling inhibitor mitigated the corticogenesis defects observed in the FTD iPSCs[56]. These findings underscore the tight associations between Wnt signalling, ALS and FTD pathogenesis. In C9ALS/FTD iPSC-derived spinal motor neurons, we observed the induction of several Wnt/β-catenin targets, namely, CCND1, FOSL1 and PITX2, at both the mRNA and protein levels (Fig. 7b–d; Supplementary Fig. 11a, b). Our findings describe Wnt/β-catenin pathway activation in C9ALS/FTD neurons.

We previously reported that excess Fuzzy protein expression is detrimental to neuronal health[8]. Fuzzy promotes the aggregation of Dishevelled (Dvl), which recruits the small GTPase Rac1[57] to activate downstream MAPK-caspase signalling and trigger apoptosis[8]. In *Fuzzy*[−/−] cells, Dvl aggregation has been shown to be inhibited; this is accompanied by an increase in evenly distributed Dvl[8], a protein conformation that promotes the activation of the Wnt/β-catenin pathway[58]. The transcription of Wnt/β-catenin target genes is known to be induced in *Fuzzy*[−/−] mouse embryos[23], suggesting a negative regulatory effect of Fuzzy on Wnt/β-catenin pathway activity. Here, we showed that Fuzzy expression was downregulated in C9ALS/FTD spinal motor neurons (Fig. 1b) and that Wnt/β-catenin pathway targets were upregulated in disease neurons (Fig. 7b, c). In particular, the upregulation of PITX2 expression intensified cell death and synaptic defects in C9ALS/FTD models (Supplementary Figs. 11h, 13a–f). Taken together, these findings suggest that tight control of Fuzzy protein expression is crucial for the maintenance of neuronal viability.

YY1 can function both as an activator and a repressor of gene transcription[32]. In this study, we demonstrated that YY1 modulates *Fuzzy* transcription through binding to two independent *Fuzzy*[YY1] sites in the *Fuzzy* promoter. Our findings illustrate that YY1 plays transcriptional activator and repressor roles at *Fuzzy*[YY1-a] and *Fuzzy*[YY1-b], respectively (Fig. 2c). Interestingly, when YY1–*Fuzzy*[YY1-a] interaction was impaired, the *Fuzzy* promoter activity was dramatically reduced (~50-fold; Fig. 2c). In contrast, only a mild induction effect (~2.5-fold) was detected when YY1–*Fuzzy*[YY1-b] interaction was impaired (Fig. 2c). This difference could be due to the fact that YY1 utilises ZF3 to bind to *Fuzzy*[YY1-a], but ZF4 to bind to *Fuzzy*[YY1-b] (Fig. 2b). Indeed, it was previously reported that mutating ZF3 resulted in a total disruption of YY1 activity, while no such effect was observed in ZF4 mutant[30]. This highlights that ZF3 plays a more critical role than ZF4 in regulating YY1 transcriptional activity.

In addition to the transcription of GGGGCC RNA from the sense strand of the pathogenic allele, the transcription of CCCCGG RNA from the antisense strand also occurs in C9ALS/FTD neurons[59]. Similar to GGGGCC RNA, the CCCCGG RNA possesses protein-binding capability towards key cellular factors such as a subset of heterogeneous nuclear ribonucleoproteins (hnRNPs)[60,61]. Additional RBPs, such as nucleolin (NCL) and DEAD-box helicase 3 X-linked, show specificity in binding to GGGGCC RNA[47,62]. In contrast, hnRNP K, a cytosine-rich RNA-binding protein, preferentially interacts with CCCCGG RNA[47]. It has been reported that GGGGCC RNA adopts G-quadruplex (G4) conformations[63,64], whereas CCCCGG RNA forms a double helix with tandem C:C mismatches[65]. In this study, we found that YY1 co-localised with GGGGCC RNA foci, but not with CCCCGG RNA foci, in C9ALS/FTD spinal motor neurons (Fig. 5a; Supplementary Fig. 15). Like YY1, NCL is a G4-binding protein[66]. We previously showed that BIND, a 21-amino acid peptide derived from the RNA recognition motif of NCL[67], can bind to GGGGCC RNA[26]. We found that BIND overexpression almost fully reversed Fuzzy downregulation and PITX2 induction in C9ALS/FTD spinal motor neurons (Fig. 8b, c). This finding further validates the role of GGGGCC RNA as the driver of downstream pathogenic YY1–Fuzzy–PITX2 signalling to induce neuronal dysfunction in C9ALS/FTD.

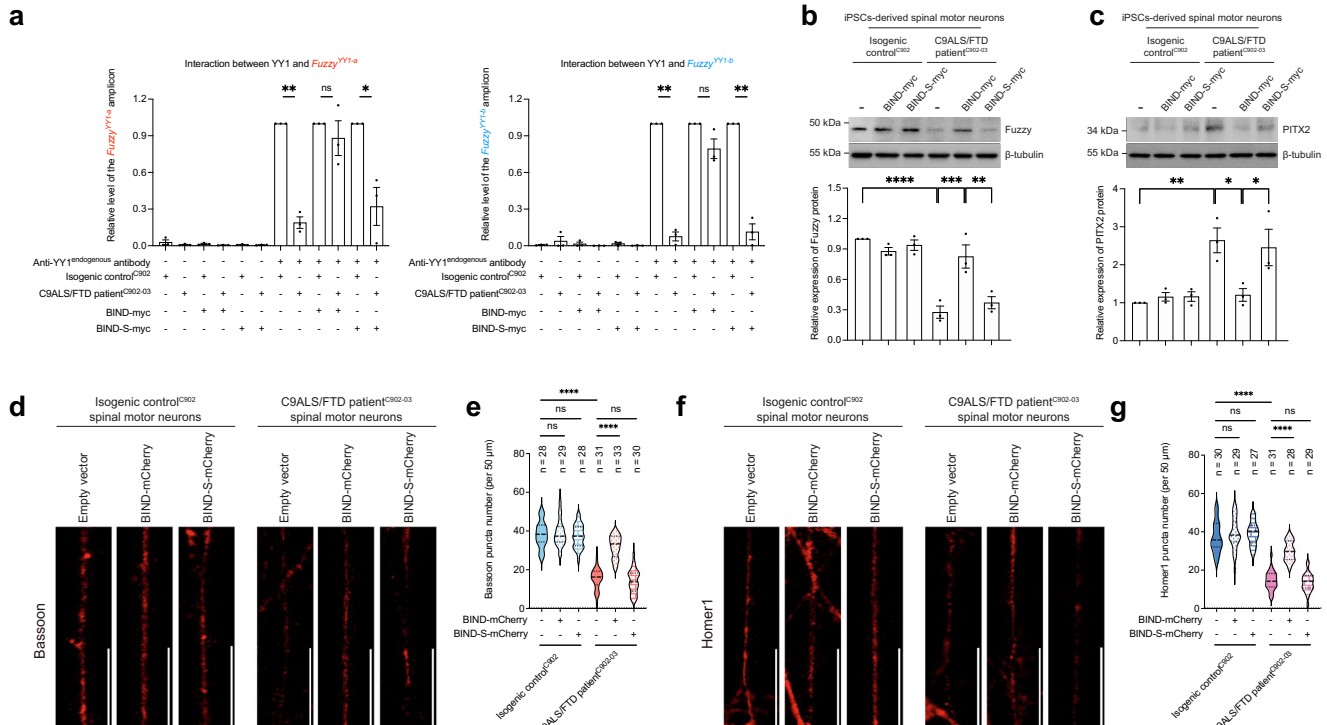

**Fig. 8 | Overexpression of BIND rescues dysregulation of YY1–Fuzzy–PITX2 signalling axis and suppresses synaptic defects in C9ALS/FTD spinal motor neurons. a** Overexpression of BIND rescued the defects in binding between YY1 and both *Fuzzy*[YY1-a] and *Fuzzy*[YY1-b] promoter sites. Over-expression of BIND-S failed to elicit a similar rescuing effect. Both *Fuzzy*[YY1-a] and *Fuzzy*[YY1-b] DNA fragments were amplified from the endogenous *Fuzzy* promoter. **b, c** Overexpression of BIND, but not BIND-S, restored the Fuzzy reduction (**b**) and PITX2 induction (**c**) in C9ALS/FTD spinal motor neurons. **d–g** Overexpression of BIND protein suppressed the reduction of Bassoon (**e**) and Homer1 (**g**) puncta numbers in C9ALS/FTD iPSC-derived spinal motor neurons, whereas BIND-S

overexpression did not cause a similar suppression effect. Overexpression of nei-ther BIND nor BIND-S affects the Bassoon (**e**) and Homer1 (**g**) puncta numbers in the isogenic control iPSC-derived spinal motor neurons. Scale bars: 20 µm. **e** and **g** are the quantifications of **d** and **f**, respectively. *n* represents the total number of neurites examined over three independent experiments. Two-tailed unpaired Student's *t*-test was used in panel (**a**). One-way ANOVA followed by *post hoc* Tukey's test was used in panels (**b**), (**c**), (**e**) and (**g**). The exact *P* values are listed in Sup-plementary Table 7. *n* = 3 biologically independent experiments and data is pre-sented as mean ± SEM. Source data are provided as a Source Data file.

In conclusion, our study revealed a pathogenic signalling cascade initiated by GGGGCC RNA that involves YY1–*Fuzzy* transcriptional dysregulation and aberrant Wnt/β-catenin pathway induction. This cascade leads to C9ALS/FTD-associated synaptic defects and cell death. An in-depth understanding of this pathogenic signalling axis could shed light on the disease mechanisms underlying C9ALS/FTD and provide new therapeutic targets to suppress C9ALS/FTD pathogenesis.

## Methods

### Ethics statement

Biological and chemical safety approval (14122815) from The Chinese University of Hong Kong was obtained for this study. All animal pro-cedures were approved by the CUHK Animal Experimentation Ethics Committee (and their care was in accordance with the institutional guidelines). All iPSC lines were derived from human skin biopsy fibroblasts, collected under ethical approval granted by the South Wales Research Ethics Committee (WA/12/0186) in the James Martin Stem Cell Facility, University of Oxford, under standardized protocols.

### Plasmids

The *pmCherry2-C1* and *pmCherry2-N1* were kind gifts from Professor Michael Davidson (Addgene plasmid # 54563 and 54517). The *pUltra* was a kind gift from Professor Malcolm Moore (Addgene plasmid # 24129)[68]. The *pMS2CP-YFP* was a kind gift from Professor Robert Singer (Addgene plasmid # 27122)[69]. The *TOPFlash-luciferase* and *FOPFlash-luciferase* were kind gifts from Professor Randall Moon (Addgene

plasmids # 12456 and 12457)[44]. The TOPFlash luciferase construct harbours 7 consecutive wildtype β-catenin binding sites, while all β-catenin binding sites in the FOPFlash luciferase construct are mutated (Fig. 7a). The *pAG3-(GGGGCC)₂* and *pAG3-(GGGGCC)₆₆* were kind gifts from Professor Leonard Petrucelli[59]. The two plasmids carry the intronic *C9orf72* sequence-appended 2 or 66 copies of GGGGCC repeats (the "GGGGCC model"; Fig. 1c). The *pcDNA3.1(+)-(GGGGCC)₃*, *pcDNA3.1(+)-(GGGGCC)₃₆*, *pcDNA3.1(+)-(GGGGCC)₉₂* and *pcDNA3.1(+)-(GGGGCC)₁₀₆-RO* were kind gifts from Professor Adrian Isaacs[39]. For *pcDNA3.1(+)-(GGGGCC)₁₀₆-RO*, a 6-bp nucleotide was inserted every 12 GGGGCC repeats to ensure no RAN translation occurs in different frames at both sense and anti-sense strands (the "GGGGCC RNA only model"; Fig. 1c). The *pcDNA3.1(+)-CCND1-flag* (Clone ID: OHu17797), *pcDNA3.1(+)-FOSL1-flag* (Clone ID: OHu10983) and *pcDNA3.1(+)-PITX2-flag* (Clone ID: OHu17360) were purchased from GenScript. The *pGL4.17[luc2/Neo]* (E6721) and *pTK-RL* (E2241) were purchased from Promega. The *phnRNP H-mCherry* was synthesized by Genewiz (Suz-hou, China). The *pcDNA3.1(+)-BIND-myc* and *pcDNA3.1(+)-BIND-S-myc* were described previously[67].

The *pGL4.17[luc2/Neo]-Fuzzy*[−2032/+574], *pGL4.17[luc2/Neo]-Fuzzy*[−1332/+574] and *pGL4.17[luc2/Neo]-Fuzzy*[−1332/+574] *YY1-b mutant* luciferase reporter constructs were described previously[8]. The *Fuzzy*[−2732/+574] DNA fragment was synthesized by GenScript and subcloned into *pGL4.17[luc2/Neo]* luciferase vector using *Nhe*I and *Bgl*II. To generate *pGL4.17[luc2/Neo]-Fuzzy*[−1142/+574] and *pGL4.17[luc2/Neo]-Fuzzy*[−632/+574], the *Fuzzy*[−1142/+574] and *Fuzzy*[−632/+574] DNA sequences were amplified from *pGL4.17[luc2/Neo]-Fuzzy*[−1332/+574] using primers *Nhe*I-*Fuzzy* *Promoter-1142-F*, 5′-CCGGCTA

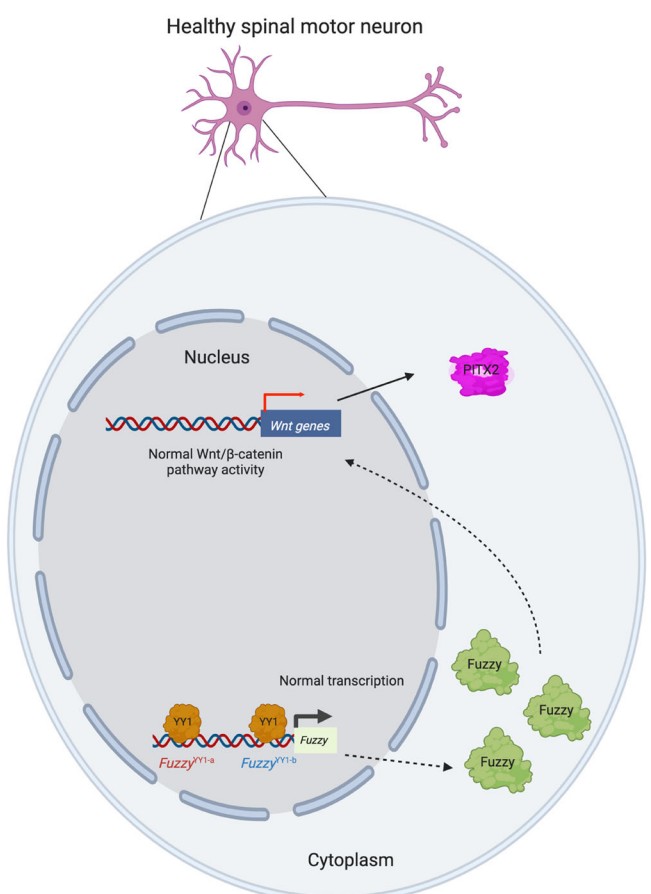
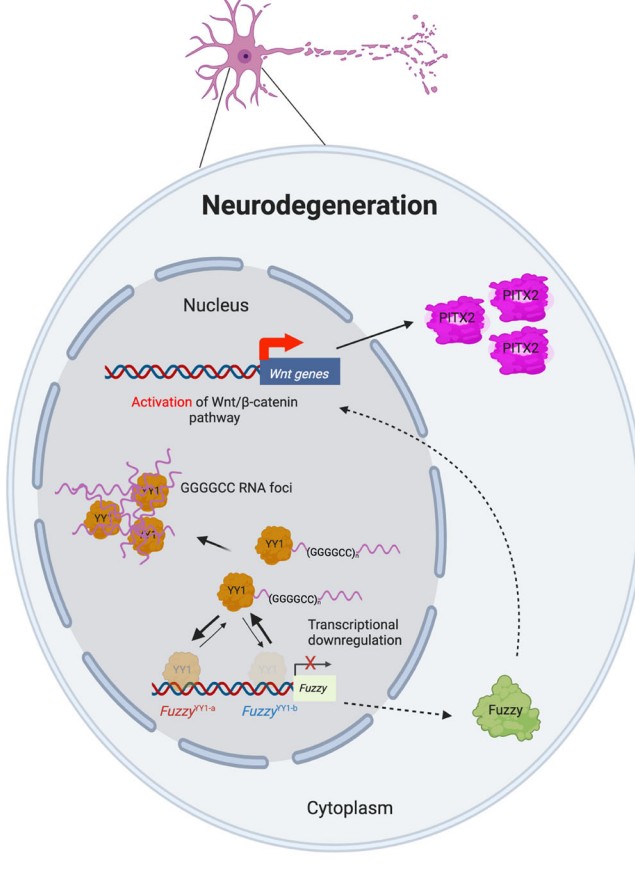

**Fig. 9 | Graphical summary of the current study.** In normal spinal motor neurons, YY1 binds to *Fuzzy* promoter at *Fuzzy*<sup>YY1-a</sup> and *Fuzzy*<sup>YY1-b</sup> sites to control the normal level of Fuzzy protein. The canonical Wnt/β-catenin pathway is not activated, and no neurodegeneration occurs. In C9ALS/FTD spinal motor neurons, YY1 binds to GGGGCC RNA. The binding of YY1 to the *Fuzzy* promoter is impaired, followed by the downregulation of the Fuzzy protein level. The Wnt/β-catenin pathway is activated. The induction of PITX2 proteins consequently contributes to the degeneration of disease neurons. The illustration was created using BioRender.com.

GCCTCCAGCCTGGGGGACGGAG-3'; *Nhe*I-*Fuzzy Promoter*-632-F, 5'-CCGGCTAGCTTTTCACATTTTACATGGTA-3' and *Bgl*II-*Fuzzy Promoter-R*, 5'-CCGAGATCTCTAAGGAGGGGTTAGGG-3'. The resultant DNA fragments were subcloned into *pGL4.17[luc2/Neo]* luciferase vector using *Nhe*I and *Bgl*II. *pGL4.17[luc2/Neo]-Fuzzy*<sup>-1332/+574</sup> *YY1-a mutant* luciferase construct was generated using site-directed mutagenesis by GenScript. To generate *pUltra-poly(GA)₅₀*, *pUltra-poly(GR)₅₀* and *pUltra-poly(PR)₅₀*, the codon-optimized *Age*I-*poly(GA)₅₀-BamHI*, *Age*I-*poly(GR)₅₀-BamHI* and *Age*I-*poly(PR)₅₀-BamHI* DNA fragments (Supplementary Table 2) were synthesized by Integrated DNA Technologies (Singapore) and then subcloned into *pUltra* vector using *Age*I and *Bam*HI.

To generate *pAG3-(GGGGCC)₂-MS2* and *pAG3-(GGGGCC)₆₆-MS2* constructs, the *Not*I-*MS2-BglII* sequence (Supplementary Table 3) was synthesized by Genewiz and then subcloned into *pAG3-(GGGGCC)₂* and *pAG3-(GGGGCC)₆₆* vectors using *Not*I and *Bgl*II. The *pYY1*<sup>full-length</sup>*-mCherry* was described previously[8]. To generate *pYY1*<sup>ZF1+ZF2+ZF3</sup>*-mCherry*, *pYY1*<sup>ZF1+ZF2</sup>*-mCherry*, *pYY1*<sup>ZF1</sup>*-mCherry* and *pYY1*<sup>ΔZF</sup>*-mCherry*, the *YY1*<sup>ZF1+ZF2+ZF3</sup>, *YY1*<sup>ZF1+ZF2</sup>, *YY1*<sup>ZF1</sup> and *YY1*<sup>ΔZF</sup> DNA sequences were amplified from *pYY1*<sup>full-length</sup>*-mCherry* using primers *Eco*RI-YY1-F, 5'-CCGGAATTC ATGGCCTCGGGCGACAC-3'; *Kpn*I-YY1<sup>1–381</sup>-R, 5'-CCGGGTACCGTC CTGTCTCCGGTATG-3'; *Kpn*I-YY1<sup>1–351</sup>-R, 5'-CCGGGTACCGTCTTCTCT CCAGTATG-3'; *Kpn*I-YY1<sup>1–323</sup>-R, 5'-CCGGGTACCGTTCTGGGACC GTGGGTG-3' and *Kpn*I-YY1<sup>1–294</sup>-R, 5'-CCGGGTACCGTTCTTGGAGCA TCATC-3'. The resultant DNA fragments were subcloned into the *pmCherry2-N1* vector using *Eco*RI and *Kpn*I. To generate *pYY1*<sup>full-length</sup>*-EGFP*, *pYY1*<sup>ZF1+ZF2+ZF3</sup>*-EGFP* and *pYY1*<sup>ZF1+ZF2</sup>*-EGFP*, the *YY1*<sup>full-length</sup>,

*YY1*<sup>ZF1+ZF2+ZF3</sup>, *YY1*<sup>ZF1+ZF2</sup> DNA fragments were subclone from *pYY1*<sup>full-length</sup>*-mCherry*, *pYY1*<sup>ZF1+ZF2+ZF3</sup>*-mCherry*, and *pYY1*<sup>ZF1+ZF2</sup>*-mCherry* to *pEGFP-N1* using *Eco*RI and *Kpn*I. To generate *pmCherry-PITX2*, the *PITX2* DNA sequence was amplified from *pcDNA3.1(+)-PITX2-flag* construct using primers *Kpn*I-PITX2-F, 5'-CCGGGTACCATGGAGACCAACTGCCGC-3' and *Bam*HI-PITX2-R, 5'-CCGGGATCCCGTCACACGGGCCGGTCCAC-3'. The resultant DNA fragment was subcloned into the *pmCherry2-C1* vector using *Kpn*I and *Bam*HI. To generate *pBIND-mCherry* and *pBIND-S-mCherry*, the double-stranded *BIND* and *BIND-S* oligonucleotides were generated using *Xho*I-BIND-*Hind*III-F, 5'-TCGAGATGGCGG AAATTCGCCTGGTGAGCAAAGATGGCAAAAGCAAAGGCATTGCGT ATATTGAATTTAAAA-3'; *Xho*I-BIND-*Hind*III-R, 5'-AGCTTTTTAAATTC AATATACGCAATGCCTTTGCTTTTGCCATCTTTGCTCACCAGG CGAATTTCCGCCATC-3'; *Xho*I-BIND-S-*Hind*III-F, 5'-TCGAGATGGGC GGCAAGATATTAAAAGCCGCGTGGAAGCGGCGAGCATTCTGTA TTTTATTAAAAAAAAAA-3' and *Xho*I-BIND-S-*Hind*III-R, 5'-AGCTTTTT TTTTTTAATAAAATACAGAATGCTCGCCGCTTCCACGCGGCTTTTAA-TATCTTGCCGCCCATC-3'. The resultant oligonucleotides were subcloned into the *pmCherry2-C1* vector using *Xho*I and *Hind*III.

**Cell culture**

The human neuroblastoma cell line SK-N-MC (HTB-10™, ATCC) was cultured using Dulbecco's Modified Eagle's Medium (SH30022.02, GE Healthcare Bio-Sciences) supplemented with 10% fetal bovine serum (F7524, Sigma-Aldrich) and 1% penicillin-streptomycin solution (15140122, Thermo Fisher Scientific). The induced pluripotent stem

cell (iPSC) lines used in this study were described previously[24,25]. The demographic information regarding these lines were summarised in Supplementary Table 1. The iPSCs were cultured on Geltrex (A1413302, Thermo Fisher Scientific) using mTeSR1 medium (85850, StemCell Technologies) supplemented with 1% penicillin-streptomycin solution. All cells were maintained in a 37 °C humidified cell culture incubator supplemented with 5% $CO_2$. All cell lines were tested negative for mycoplasma contamination using the MycoAlert™ plus Mycoplasma Detection Kit (LT07-703, Lonza). Despite being listed as a misidentified cell line (sarcoma instead of neuroblastoma; Registration ID: ICLAC-00267), our SK-N-MC cells are obtained from a different commercial source (ATCC, USA) as the previously misidentified SK-N-MC cells (Deutsche Sammlung für Zellkulturen und Mikroorganismen; Germany)[70]. Our SK-N-MC cell line is of neurogenic origin as stated by ATCC (https://www.atcc.org/products/htb-10). No additional authentication of cell lines was performed in our laboratory.

## Spinal motor neuron differentiation
The spinal motor neuron differentiation was performed as described previously[24], with minor modifications. Briefly, on day 0 to day 1, iPSCs were cultured in basal medium containing DMEM/F12 (11320033, Thermo Fisher Scientific)/Neurobasal (12348017, Thermo Fisher Scientific) 1:1, 1x B27 supplement (17504001, Thermo Fisher Scientific), 1x N2 supplement (17502001, Thermo Fisher Scientific), 1x penicillin-streptomycin solution, 1x β-mercaptoethanol (21985023, Thermo Fisher Scientific), 0.5 μM L-ascorbic acid (A92902, Sigma-Aldrich) supplemented with 3 μM Chir99021 (72052, StemCell Technologies) and 1 μM compound C (100-0246, StemCell Technologies). On day 2, the medium was supplemented with 1 μM retinoic acid (R2625, Sigma-Aldrich) and 0.5 μM Smoothened agonist (4366/1, Tocris Bioscience). On day 4, Chir99021 and compound C were removed from the medium. On day 9, neural precursors were split at 1:3 using StemPro™ Accutase™ (A1110501, Thermo Fisher Scientific), and 10 μM Rock inhibitor (72302, StemCell Technologies) was included for 24 h. The neural precursors were maintained in the same medium until day 16 when the cells were split again and plated at desired densities in culture plates. The medium was changed to basal medium supplemented with 1 μM retinoic acid, 0.5 μM Smoothened agonist, 0.5 μg/ml laminin (L2020, Sigma-Aldrich), 10 ng/ml human brain-derived neurotrophic factor recombinant protein (PHC7074, Thermo Fisher Scientific), 10 ng/ml human glial cell line-derived neurotrophic factor recombinant protein (PHC7045, Thermo Fisher Scientific). Rock inhibitor, Cytarabine (2 μM, 4520/50, Tocris Bioscience), and DAPT (10 μM, 2634/10, Tocris Bioscience) were added for 2, 3 and 7 days and then withdrawn for the rest of neural maturation. Experiments were performed using day 30 neurons in all experiments. A detailed differentiation scheme is presented in Supplementary Fig. 1a.

## Plasmid and siRNA transfection
The SK-N-MC cells were transfected with plasmids and siRNAs using Lipofectamine™ 2000 (11668019, Thermo Fisher Scientific) and Lipofectamine™ RNAiMAX (13778150, Thermo Fisher Scientific), respectively. The transfection was carried out according to the manufacturer's instructions. The ON-TARGETplus SMARTpool siRNAs from Dharmacon (Horizon Discovery Ltd.) were used to knock down the expression of *YY1*, *C9orf72*, *CCND1*, and *FOSL1*. Specifically, 5 pmol of *C9orf72*-siRNA (L-013341-01-0005), 5 pmol of *YY1*-siRNA (L-011796-00-0005), 5 pmol of *CCND1*-siRNA (L-003210-00-0005), and 5 pmol of *FOSL1*-siRNA (L-004341-00-0005) were used. Non-targeting siRNA (D-001210-01-50) was used as a control. The *PITX2*-siRNA (CCCAGG-CUAUUCCUACAACAAUT) was purchased from Sangon Biotech (Shanghai, China) and used at 5 pmol to knock down *PITX2* expression. The non-targeting siRNA (UUCUCCGAACGUGUCACGUTT) was used as the control. Gene knockdown was performed twice.

The iPSC-derived spinal motor neurons were transfected with *pcDNA3.1(+)-BIND-myc*, *pcDNA3.1(+)-BIND-S-myc*, *pBIND-mCherry*, or *pBIND-S-mCherry* plasmid on day 24 post differentiation (Supplementary Fig. 1a). Lipofectamine™ Stem transfection reagent (STEM00003, Thermo Fisher Scientific) was used, and the transfection was performed according to the manufacturer's instructions. Lipofectamine™ RNAi-MAX was used for the siRNA transfection. The 20 pmol of *PITX2*-siRNA (Sangon Biotech) was used. The transfection was performed twice on day 24 and day 27 post differentiation (Supplementary Fig. 1a).

## Peptide treatment
The TAT-BIND (YGRKKRRQRRRAEIRLVSKDGKSKGIAYIEFK) peptide (synthesized and HPLC-purified by GenScript) was used to treated SK-N-MC cells, and the treatment lasted 24 h.

## Reverse transcription (RT)-PCR
The RNA samples were prepared from SK-N-MC cells, iPSC-derived spinal motor neurons, and fly heads using TRIzol™ reagent (15596018, Thermo Fisher Scientific). The RT was performed using ImProm-II™ Reverse Transcription System (A3803, Promega) following the manufacturer's instructions. Oligo dT primer (5'-TTTTTTTTTTTTTTTT-3') was used for the RT. Primers used for conventional PCR are summarised in Supplementary Table 5. The PCR parameters used were: after the initial step at 95 °C for 5 min, the products were denatured at 95 °C for 30 s, followed by annealing at 55 °C for 30 s, and extension at 72 °C for 20 s. The denaturation-annealing-extension cycle was repeated 24 times (for *β-actin*, product size: 244 bp), 26 times (for *Pho*, product size: 205 bp), 28 times (for *Ptx1*, product size: 371 bp), 26 times (for *GGGGCC*, product size: 113 bp) or 26 times (for *YY1*, product size: 337 bp). The products were then undergone final extension at 72 °C for 10 min before being subjected to agarose gel electrophoresis. The images were captured and processed using ChemiDoc™ Touch Imaging System (170-8370, Bio-Rad Laboratories). *β-actin* was used as the loading control. Only representative gels are shown.

## Real-time PCR (qPCR)
The human Wnt signalling pathway RT² Profiler PCR Array (PAHS-043YF, Qiagen) was used to determine levels of Wnt-responsive genes in C9ALS/FTD and control iPSC-derived spinal motor neurons. Each RT product was mixed with RT² SYBR Green qPCR Mastermix (330503, Qiagen) and aliquoted into the PCR arrays. Thermal cycling was performed on Roche LightCycler® 480 System under the following conditions: 10 min at 95 °C, 45 cycles of 15 s at 95 °C and 1 min at 60 °C. The Ct values of the internal control genes (*ACTB*, *B2M*, *GAPDH*, *HPRT1*, *RPLPO*) were averaged ($Ct_{internal\ control}$), and relative gene expression was quantified using the $2^{-\Delta\Delta Ct}$ method, where $\Delta\Delta Ct = (Ct_{target\ gene} - Ct_{internal\ control})_{C9ALS/FTD} - (Ct_{target\ gene} - Ct_{internal\ control})_{control}$.

For the qPCR performed in chromatin immunoprecipitation and Click-iT nascent RNA capture assays, SYBR™ Green PCR Master Mix (4364346, Thermo Fisher Scientific) was used. Thermal cycling was performed on Bio-Rad CFX96 Real-time PCR detection system (Bio-Rad Laboratories) under the following conditions: 10 min at 95 °C, and 50 cycles of 15 s at 95 °C and 1 min at 60 °C. Primers used for qPCR are summarised in Supplementary Table 5.

## Chromatin immunoprecipitation (ChIP)
The ChIP assay was performed using Pierce™ Magnetic ChIP Kit (26157, Thermo Fisher Scientific). The experimental procedures were carried out following the manufacturer's instructions. Four micrograms of anti-YY1 (22156-1-AP, Proteintech) or anti-mCherry (ab167453, Abcam) antibody was used for the IP, while the same amount of normal rabbit IgG (I-1000, Vector Laboratories, Inc.) was used as a negative control. The anti-YY1 (1:1000, MAB3784, R&D Systems, Inc.) and anti-mCherry (1:1000, NBP1-96752, Novus Biologicals) antibodies were used for the

detection of the endogenous YY1 protein and YY1-mCherry proteins in the input and IP samples. Twenty nanograms of recovered genomic DNAs from each input, normal rabbit IgG, and YY1 or mCherry immunoprecipitated samples were used in the following qPCR to analyse the levels of *Fuzzy* promoter fragments. The primers used were *Fuzzy*$^{YY1-a}$-F, 5′-GGTGGGTGGATCATTTGAGC-3′; *Fuzzy*$^{YY1-a}$-R, 5′-TTTTG GGAGAGGGGACAGG-3′; *Fuzzy*$^{YY1-b}$-F, 5′-GACCACAGGCCGAGTTACTC-3′ and *Fuzzy*$^{YY1-b}$-R, 5′-TTCGGTTTAGATACAGGCGTC-3′. Relative gene expression in IgG IP samples was quantified using the $2^{-\Delta Ct}$ method, where $\Delta Ct = Ct_{IgG\ IP} - Ct_{Input}$. Relative gene expression in antibody IP samples was quantified using the $2^{-\Delta\Delta Ct}$ method, where $\Delta\Delta Ct = (Ct_{antibody\ IP} - Ct_{Input})_{disease} - (Ct_{antibody\ IP} - Ct_{Input})_{control}$.

**Click-iT nascent RNA capture assay**

The Click-iT nascent RNA capture assay (C10365, Thermo Fisher Scientific) was performed following the manufacturer's instructions. The disease and control iPSC-derived spinal motor neurons differentiated for 30 days were labelled with 0.5 mM 5-ethynyl uridine (EU) at 37 °C with 5% CO$_2$ for 1 h. The total RNA was subsequently harvested using RNeasy Mini Kit (74104, Qiagen) supplemented with DNase treatment (79254, Qiagen). The biotin-azide was conjugated to EU to generate the biotinylated RNA. For each sample, 12 μL of Streptavidin magnetic beads were mixed with 200 ng biotinylated RNA to capture EU-labelled nascent transcripts. The on-beads cDNA synthesis was performed using SuperScript VILO™ cDNA synthesis kit (11754050, Thermo Fisher Scientific), followed by the qPCR to determine levels of nascent *Fuzzy* and *YY1* transcripts using primers listed in Supplementary Table 5.

**Immunocytochemistry (ICC)**

For ICC performed on iPSC-derived spinal motor neurons, neurons were fixed with 4% paraformaldehyde for 15 min, followed by permeabilization with 0.2% Triton X-100 for another 15 min. The cells were then blocked with 5% donkey serum at 25 °C for 1 h before being incubated with primary antibodies. The primary antibodies used were anti-PITX2 (1:200, H00005308-M01, Novus Biologicals), anti-Tau (1:500, ab75714, Abcam), anti-Bassoon (1:200, ab110426, Abcam), anti-MAP2 (1:1000, ab5392, Abcam), and anti-Homer1 (1:200, ab97593, Abcam). Secondary antibodies used in this study were Alexa Fluor 488 Goat anti-Chicken IgY H&L (1:500, ab150169, Abcam), Alexa Fluor 594 Donkey anti-Mouse IgG (H + L) (1:500, A-21203, Thermo Fisher Scientific), and Alexa Fluor 647 Donkey anti-Rabbit IgG (H + L) (1:500, A-31573, Thermo Fisher Scientific). The number of Bassoon or Homer1 puncta was counted alongside each neurite of 50 μm in length. The cell nuclei were stained with Hoechst 33342 (1:400, H-1399, Thermo Fisher Scientific). Cell images were acquired on an Olympus IX-81 FV1000 confocal microscope (Olympus) using a 63x water immersion objective lens and Olympus Fluoview software (Version 4.2a). Only representative images are shown.

**Fluorescence in situ hybridisation coupled with ICC**

Forty-eight hours post transfection, cells were fixed using 4% paraformaldehyde for 15 min and permeabilized with 0.2% Triton X-100 at 25 °C for 10 min. Cells were then washed with diethylpyrocarbonate-treated PBS (DEPC-PBS) twice before being incubated with pre-hybridisation buffer (40% formamide, 2X Saline Sodium Citrate (SSC)) at 65 °C for 30 min. Cells were then hybridized with: (1) denatured TYE563-labelled locked nucleic acid probe 5′-TYE563-CCCGGCCCCGGCCCC-3′ (40 nM, Exiqon) for the visualization of GGGGCC RNA foci in SK-N-MC cells[26]; or (2) denatured Cy3-conjugated 2′O-methyl RNA probe 5′-Cy3-mCmCmCmCmGmGmCmCmCmCmGmGmCmCmCmCmGmGmC-mCmCmCmGmG-3′ (80 nM, Integrated DNA Technologies) for the visualization of GGGGCC RNA foci in iPSC-derived spinal motor neurons[71]; or 3) denatured TYE563-labelled locked nucleic acid

probe 5′-TYE563-GGGGCCGGGGCCGGG-3′ (80 nM, Exiqon) for the visualization of CCCCGG RNA foci in iPSC-derived spinal motor neurons. The cells or neurons were incubated in hybridisation buffer (40% formamide, 2X SSC, 1 mg/mL salmon sperm DNA, 40 U RNAseOUT, 0.1 mg/mL tRNA, 10% dextran sulphate, 2 mM vanadyl ribonucleoside) at 65 °C for 4 h. After washing three times with pre-hybridisation buffer at 65 °C each for 10 min, cells were further washed with 2X SSC three times at 25 °C each for 10 min.

The ICC was subsequently performed. Primary antibodies used were anti-YY1 (1:200, ab109237, Abcam), anti-YY1 (1:200, 22156-1-AP, Proteintech), anti-hnRNP H (1:200, ab10374, Abcam), anti-GFP (1:500, 632381, Takara Bio Inc.), and anti-Tuj1 (1:1000, 801202, BioLegend). Secondary antibodies used were Alexa Fluor 488 Donkey anti-Mouse IgG (H + L) (1:500, A-21202, Thermo Fisher Scientific) and Alexa Fluor 647 Donkey anti-Rabbit IgG (H + L) (1:500, A-31573, Thermo Fisher Scientific). The cell nuclei were stained with Hoechst 33342. Cell images were acquired on an Olympus IX-81 FV1000 confocal microscope using a 63x water immersion objective lens and Olympus Fluoview software. Only representative images are shown.

**Fluorescence recovery after photobleaching (FRAP)**

The SK-N-MC cells were seeded onto a 35-mm coverglass bottom confocal dish (SPL-100350, SPL Life Sciences Co., Ltd.). Twenty-four hours later, the cells were transfected with 1 μg of *pAG3-(GGGGCC)$_{2/66}$-MS2*, 0.5 μg of *pMS2CP-YFP*, together with 0.2 μg of *phnRNP H-mCherry* or 0.5 μg of *pYY1-mCherry* construct for 48 hours. In the FRAP experiment, cells were maintained in DMEM without phenol red (31053028, Thermo Fisher Scientific). Region of interest (ROI) was selected and photobleached using the FRAP module on a TCS SP8 high speed imaging system (Leica Microsystems). All pre-bleach, photobleaching, and post-bleach steps were controlled automatically by the "Zoom in" mode of Leica Application Suite X software. Cells were imaged before bleach (pre-bleach), followed by the bleach with 2 times 100% power of the 561-nm laser, and post-bleach for twenty iterations. Only representative images are shown. The fluorescent intensity of the bleached ROI at each timepoint was automatically calculated on the Leica Application Suite X software. These fluorescent intensities were normalized against the fluorescent intensity from the pre-bleach ROI to determine the fluorescent recovery.

**Immunoblotting**

Protein samples were harvested from SK-N-MC cells or iPSC-derived spinal motor neurons using the SDS sample buffer (100 mM Tris-HCl, pH 6.8, 2% SDS, 40% glycerol, 5% β-mercaptoethanol, and 0.1% bromophenol blue). Samples were heated at 99 °C for 10 min before being subjected to the immunoblotting analysis. The protein samples were then transferred to a PVDF membrane (IPVH00010, pore size 0.45 μm, Merck Millipore). The membrane was blocked using 5% non-fat milk at 25 °C for 1 h, followed by incubating primary antibodies at 4 °C for 16 h. Primary antibodies used in this study were anti-Fuzzy (1:1000, ab111842), anti-YY1 (1:1000, ab109237), anti-CCND1 (1:1000, ab134175), anti-FOSL1 (1:1000, ab124722), anti-PITX2 (1:1000, ab32832), anti-β-tubulin (1:2000, ab6046) from Abcam, and anti-C9orf72 (1:1000, GTX632041) from Genetex (Irvine, USA).

The membrane was washed three times with 1× TBST each for 10 min before being subjected to the incubation of secondary antibodies at 25 °C for 1 h. Secondary antibodies used were HRP-conjugated goat anti-rabbit IgG (H + L) (11-035-045, 1:5000) and HRP-conjugated goat anti-mouse IgG (H + L) (115-035-062, 1:10,000) from Jackson ImmunoResearch (West Grove, USA). The membrane was washed three times with 1× TBST each for 10 min, followed by chemiluminescent signal detection. The signal was developed using Immobilon Forte Western HRP substrate (WBLUF0100, Merck Millipore), and the images were captured and processed using ChemiDoc™

Touch Imaging System. β-tubulin was used as the loading control. Only representative blots are shown.

### Dual-luciferase reporter assay

Dual-luciferase reporter assay was performed according to the manufacturer's instructions (E1910, Promega). The readings of firefly and *Renilla* luminescence were recorded on a FLUOstar Omega Microplate Reader (BMG LABTECH, Germany). The "Relative promoter activity" denotes the normalization of readings generated from firefly luciferase construct (*pGL4.17[luc2/Neo]-Fuzzy*$^{-2732/+574}$, *pGL4.17[luc2/Neo]-Fuzzy*$^{-2032/+574}$, *pGL4.17[luc2/Neo]-Fuzzy*$^{-1332/+574}$, *pGL4.17[luc2/Neo]-Fuzzy*$^{-632/+574}$, *pGL4.17[luc2/Neo]-Fuzzy*$^{-1142/+574}$, *pGL4.17[luc2/Neo]-Fuzzy*$^{-1332/+574}$ YY1-a mutant, *pGL4.17[luc2/Neo]-Fuzzy*$^{-1332/+574}$ YY1-b mutant, TOP Flash or FOP Flash) against the readings generated from *Renilla* luciferase construct (*pRL-TK*).

### Lactate dehydrogenase (LDH) cytotoxicity assay

The LDH cytotoxicity assay was performed using a CytoTox 96® non-radioactive cytotoxicity assay kit (G1780, Promega) according to the manufacturer's instructions. Briefly, SK-N-MC cells were cultured in 24-well plates, after transfection, the cells were applied for the cytotoxicity assay. Both the absorbance of LDH enzyme released into the culture medium ($Absorbance_{supernatant}$) and from attached cells ($Absorbance_{attached}$) were obtained on a Spark® multimode microplate reader (Tecan, Switzerland). The relative cell death was calculated using the following equation and the data from experimental groups were normalized to respective controls.

$$\text{Relative cell death} = \frac{Absorbance_{supernatant}}{Absorbance_{supernatant} + Absorbance_{attached}}$$

### Electrophoretic mobility shift assay

Both DNA and RNA oligonucleotides (Supplementary Table 4) were synthesized by Integrated DNA Technologies and Dharmacon and dissolved in buffer containing 8 mM Na$_2$HPO$_4$, 185 mM NaCl and 1 mM EDTA. The DNA and RNA oligonucleotides were denatured at 60 °C for 8 min, refolded at 37 °C for 30 min, and then gradually cooled down to 25 °C prior to use. The *(GGGGCC)$_8$* RNA and randomised GC-rich RNA were denatured at 95 °C for 5 min and then gradually cooled down to 25 °C prior to use. The recombinant His-YY1$^{full-length}$, His-YY1$^{ZF1+ZF2+ZF3}$ and His-YY1$^{ZF1+ZF2}$ proteins were produced by GenScript and dissolved in buffer containing 50 mM Tris-HCl, 500 mM NaCl, 10% Glycerol, pH 8.0. The denatured DNA or RNA oligonucleotide (2.5 nM of each) was mixed with different concentrations of recombinant YY1 proteins and 20 U RNaseOUT recombinant ribonuclease inhibitor (10777019, Thermo Fisher Scientific) in binding buffer (10 mM HEPES-KOH, 12 mM Tris-HCl, 50 mM NaCl, 50 mM KCl, 5 mM MgCl$_2$, 100 μM ZnCl$_2$, 0.01% NP-40, 5% glycerol, 2.5 mM DTT, pH 7.4). The mixtures were incubated at 25 °C in dark for 30 min before being loaded onto 6% nondenaturing polyacrylamide gels for electrophoresis on ice in 0.5x TB running buffer (45 mM boric acid, 45 mM Tris, 50 μM ZnCl$_2$).

For the competition assays, 2.5 nM of Cy5-*Fuzzy*$^{YY1-a}$ or Cy5-*Fuzzy*$^{YY1-b}$ DNA was mixed with 2.65 mM recombinant His-YY1$^{full-length}$ protein (ab187479, Abcam) and 20 U RNaseOUT recombinant ribonuclease inhibitor in binding buffer. An increasing amount of *(GGGGCC)$_8$* RNA (5′-GGGGCCGGGGCCGGGGCCGGGGCCGGGGCCGGGGCCGGGGCCGG GGCC-3′) was introduced to the YY1-*Fuzzy*$^{YY1}$ DNA binding reactions. The mixtures were incubated at 25 °C in dark for 30 min and then loaded onto 6% nondenaturing polyacrylamide gels for electrophoresis on ice in 0.5x TB running buffer. The gels were visualized, and images were captured using ChemiDoc$^{TM}$ MP System (17001402, Bio-Rad). Only representative gels are shown. The band intensities of both free DNA/RNA probes ($Intensity_{free\ probe}$) and YY1-nucleotide complexes that are

resolved in the gel ($Intensity_{complex}$) were quantified using the ImageJ software. The percentage of shifted DNA/RNA signals was calculated by the following equation: DNA/RNA fraction bound (%) = $Intensity_{complex}$ / ($Intensity_{free\ probe}$ + $Intensity_{complex}$) × 100%. The obtained values were used to plot the YY1-RNA binding curves. The binding affinity ($K_d$) was determined using the non-linear regression model on the GraphPad Prism software (version 9.2.0).

### Fluorescence polarisation (FP) assay

The FAM-labelled *(GGGGCC)$_8$* RNA oligonucleotide (Supplementary Table 4) was dissolved in buffer containing 8 mM Na$_2$HPO$_4$, 185 mM NaCl and 1 mM EDTA. The RNA oligonucleotide was denatured at 60 °C for 8 min, refolded at 37 °C for 30 min, and then gradually cooled down to 25 °C prior to use. The denatured RNA oligonucleotide (2.5 nM) was mixed with different concentrations of recombinant His-YY1$^{full-length}$ protein in binding buffer (50 mM Tris-HCl, 100 mM KCl, 2.5 mM MgCl$_2$, 100 μM ZnCl$_2$, pH 7.4). The mixture was incubated at 25 °C in dark for 30 min. The fluorescence polarisation was measured on a Spark® multimode microplate reader with the excitation and emission wavelengths of 485 nm and 535 nm, respectively. The parallel ($I_{parallel}$) and perpendicular ($I_{perpendicular}$) fluorescence intensities were determined by the SparkControl software (version 2.1), following by the calculation of millipolarisation (mP) units = 1000 × ($I_{parallel}$ − $I_{perpendicular}$) / ($I_{parallel}$ + $I_{perpendicular}$). The binding affinity ($K_d$) was determined using the non-linear regression model on the GraphPad Prism software (version 9.2.0).

### *Drosophila* genetics

All *Drosophila melanogaster* stocks and genetic crosses were maintained on the cornmeal medium in a 25 °C incubator. The *elav$^{GS}$-Gal4* (43642), *UAS-(GGGGCC)$_3$* (58687), and *UAS-(GGGGCC)$_{36}$* (58688) lines were obtained from Bloomington *Drosophila* Stock Center (Bloomington, USA). The *UAS-Ptx1-dsRNA$^{KKI08574}$* (107785) line was obtained from Vienna *Drosophila* RNAi Center (Vienna, Austria). The *UAS-Pho* (F000151) and *UAS-Ptx1* (F003469) lines were obtained from FlyORF (Zurich, Switzerland). Both male and female flies were used in all experiments.

### *Drosophila* climbing and survival assays

The 2 days post eclosion (dpe) flies were transferred to fly food containing 200 μM mifepristone (RU486, ab120356, Abcam). The RU486 was used to induce transgene expression. The climbing ability of flies was measured by negative geotaxis assay. Flies from each experimental group were anesthetized and placed in a vertical plastic column. After an hour of recovery in 25 °C incubator, flies were banged to the bottom, and the time they used to climb across 6 cm in 30 s was recorded. For those who failed to climb across 6 cm in 30 s, the climbing time was marked as 30 s. The ages of flies used are: 14 dpe (Figs. 6d and 7f) and 10 dpe (Supplementary Fig. 12c). For the survival assay, the survival rate of flies was recorded daily until flies from each experimental group were all dead. The survival plots were generated by using the GraphPad Prism software (version 9.2.0).

### Analysis of public RNA-seq datasets

The raw sequencing data from public RNA-seq datasets were downloaded from Gene Expression Omnibus (GEO) with the accession codes GSE143743 and GSE168831. The datasets were imported to Partek Flow Genomic Analysis Software for analysis. The reads were aligned to the hg38 build of the human genome under the "STAR" mode. The aligned reads were further quantified to Ensembl Transcripts release 105 annotation model to generate gene counts. After filtering out genes with low read counts, the filtered counts among different samples were normalized using the "Median ratio" method. The DESeq2 method was subsequently used for the analysis of normalized counts to obtain differentially expressed genes. The lists of differentially expressed

genes from NeuroLINCS dataset and AnswerALS data portal were downloaded directly from the published paper[72].

## Software

The intensity of the protein or DNA bands were quantified using the ImageJ software (https://imagej.nih.gov/ij/)[73]. The differential expressed gene lists were generated using the Partek Flow Genomic Analysis Software (https://www.partek.com/partek-flow/). The illustrations were created using Adobe Illustrator 27.0.1 and BioRender.com. Assembly of panels in finalized figures was done using Adobe Illustrator 27.0.1.

## Statistical analysis

The two-tailed, unpaired Student's *t* test or Mann–Whitney *U* test was used for the comparison between two experimental groups. One-way ANOVA followed by *post hoc* Tukey's test or one-way ANOVA followed by *post hoc* Dunnett's test was applied when comparing three or more groups. Log-rank (Mantel-Cox) test was used for survival analysis. *, **, ***, and **** represent $P < 0.05$, $P < 0.01$, $P < 0.001$, and $P < 0.0001$, respectively, which are considered statistically significant. ns indicates no significant difference. The violin plots show the median (dashed lines), and 25th and 75th quartiles (dotted lines). GraphPad Prism version 9.2.0 was used to plot data and perform statistical analysis.

## Reporting summary

Further information on research design is available in the Nature Portfolio Reporting Summary linked to this article.

## Data availability

Transcription factor binding sites were predicted using Promo (http://alggen.lsi.upc.es/cgi-bin/promo_v3/promo/promoinit.cgi?dirDB=TF_8.3)[74], JASPAR (http://jaspar.genereg.net/)[75] and Animal TFDB 3.0 (http://bioinfo.life.hust.edu.cn/AnimalTFDB/#!/)[76] databases. The *Fuzzy* promoter sequence was withdrawn from GenBank under the accession number NG_032843.1. The YY1 zinc finger domain amino acid sequences were withdrawn from GenBank under the accession numbers NP_003394.1 (human), XP_009426747.2 (chimpanzee), NP_001091550.1 (cattle), XP_020955358.1 (pig), NP_775412.1 (rat), NP_033563.2 (mouse), NP_001116880.1 (frog) and NP_524630.1 (fruit fly). The public transcriptomic datasets used in this study are available in the GEO under accession codes GSE143743 and GSE168831. The data supporting the findings of this study are available from the corresponding authors upon request. Source data are provided with this paper.

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

## Acknowledgements

We thank the former and current members from Laboratory of *Drosophila* Research and Oxford Motor. Neuron Disease Centre for their insightful comments. The *pAG3-(GGGGCC)$_2$* and *pAG3-(GGGGCC)$_{66}$* plasmids were kind gifts from Professor Leonard Petrucelli (Mayo Clinic). The *pcDNA3.1(+)-(GGGGCC)$_3$*, *pcDNA3.1(+)-(GGGGCC)$_{36}$*, *pcDNA3.1(+)-(GGGGCC)$_{92}$* and *pcDNA3.1(+)-(GGGGCC)$_{106}$-RO* plasmids were kind gifts from Professor Adrian Isaacs (University College London). This work was supported by CUHK Gerald Choa Neuroscience Institute and CUHK Direct Grant for Research (4053542; to H.Y.E.C.), and a Postdoctoral Fellowship in Clinical Neurosciences program between The Chinese University of Hong Kong and University of Oxford (Nuffield Department of Clinical Neurosciences and Pembroke College, University of Oxford; to Z.S.C.).

## Author contributions

Z.S.C., K.T. and H.Y.E.C. planned the study and designed the experiments. Z.S.C., M.O., S.T. and R.D. carried out the experiments and analyses. Z.S.C., M.O., S.T. and S.I.P. interpreted the results. Z.S.C. and H.Y.E.C. wrote and revised the manuscript.

## Competing interests

The authors declare no competing interests.
