## [Peer Review File · Nature Communications]

Mutant GGGGCC RNA prevents YY1 from binding to Fuzzy promoter which stimulates Wnt/ β -catenin pathway in C9ALS/FTDREVIEWER COMMENTS

Reviewer #1 (Remarks to the Author):

An expansion of GGGGCC repeats within the C9ORF72 is the first genetic cause of Amyotrophic Lateral Sclerosis the 3rd most common neurodegenerative disease worldwide. How these repeats drive neuronal cell degeneration is still unclear. In this work, Chen and collaborator found that the YY1 transcription factor binds to and is titrated off its natural targets, notably binding to the Fuzzy gene promoter, ultimately resulting in mis-regulation of the Wnt/B-catenin pathway. Overall, Chen et al., findings are novel and provocative, relatively well controlled and well thought. However, this work raises several questions and require various additional experiments before to be considered for publication.

- Fig.1. How Fuzzy mis-regulation in C9 iPS cells was found to stand out from other genes/candidates is unclear. RNA seq / screen or any other entry point showing Fuzzy mis-regulation as a solid candidate compared to other mis-regulations should be shown. Moreover, Fig. 1a/b show results and quantifications from classical RT-PCR and absolutely need to be confirmed by much quantitative assays (RNA Seq or real time qPCR).

- Fig. 2. YY1 binding and activity on Fuzzy promoter and expression are relatively well controlled. However, binding of YY1 to GGGGCC RNA requires additional evidences, notably comparison with GGGGCC DNA and other RNA and DNA sequences, especially ones previously found to bind to YY1 as + controls. As negative controls, binding to randomized G4 C2 RNA mutants should be tested. Kd should be indicated.

- Fig. 3. YY1 co-localization within GGGGCC RNA foci should be confirmed with another antibody or with Fluorescent Protein (FP)-tagged YY1. Titration/ sequestration and/or at least decreased mobility of FP-YY1 need to be tested by FRAP analysis and/or Dendra conversion (PS, an image of co-localization is not evidence of immobilization or sequestration, namely YY1 could be enriched within GGGGCC RNA foci but still able to move in and out). Eventually, comparison with a positive control (hnRNPH or hnRNPF) could be presented. Similarly, a negative control including a RNAase treatment would be a nice addition.

IPS / motor neurons cell experiments sound well performed and well controlled, but as non-expert in such models, other Referee opinions will be required.

Reviewer #2 (Remarks to the Author):

A (G4C2)_n expansion in C9orf72 is the most common genetic cause of amyotrophic lateral sclerosis (ALS) and frontotemporal dementia (FTD). Repeat RNA gain of toxicity is recognized as one of the primary disease mechanisms. In this manuscript by Chen et al, the authors reported that the zinc finger (ZF) transcription regulator YY1 binds to GGGGCC repeat RNA and this interaction compromises the binding of YY1 to the Fuzzy promoter site, resulting in downregulation of Fuzzy. This consequently activates the Wnt/ β -catenin pathway, which induces synaptic deficits in C9ALS/FTD neurons. This study proposes a novel mechanism of repeat RNA mediated toxicity. However, more evidence is needed to fully support the proposed model.

Major concerns:

1. As the GGGGCC RNA can interfere with the binding of YY1 on both Fuzzy YY1-a and YY1-b regions, even bigger effect on 1-b which mediates the suppression of the promoter, why the RNA leads to downregulation of the targets predominantly? In Figure 2h, why repeat expression has no effect on the YY1-a mutant?
2. In Figure 2b, YY1 knockdown should be performed to show the effect is mediated through YY1.
3. The binding of YY1 on the repeat RNA needs more convincing evidence. In Figure 3a, the colocalization of YY1 with RNA foci is not obvious. The staining of YY1 in GGGGCC66 cells has no significant difference from the GGGGCC2 cells. And in all the YY1 staining data including reporter cells and iPSC-motor neurons, only one image of cells was shown. There is no quantification or statistical analysis of the differences. In Figure 3b, the difference between GGGGCC2 and GGGGCC66 is surprisingly small. This is hard to explain the specific effect on the function of YY1 by the GGGGCC66. The low amount of binding will not sequester sufficient proteins to influence the overall function.
4. In Figure 3c,d, as the colocalization assay cannot show convincing data, an alternative method is needed to show the binding and the domain required for the binding. The gel shift assay could help.
5. Figure 3e, the two C9 lines are not consistent. And again there is no statistics. Only two C9 iPSC lines were used in the whole manuscript. The number is low considering the variation of individual human lines.
6. The authors should perform ChIP for the endogenous promoter in iPSC-differentiated motor neurons comparing control and C9 lines. And the transcription activity of the endogenous YY1 target should be measured (such as by Click-iT Nascent RNA capture) to show the transcription is changed in patient neurons. The same assays should be performed in cells overexpressing YY1 to show the rescue at the molecular signatures besides synaptic phenotypes.
7. In Figure 4e, the GGGGCC repeat RNA levels should be measured to confirm the RNA expression is not altered due to cotransfection. This will exclude the possibility that the reduced cell death is due to the decreased repeat RNA level.
8. Many important details are missing in the writing. It is hard to read and get the information how each experiment was done. For example, how the luciferase data is normalized? In the method, it is

mentioned dual luciferase. But the labeling in the figures showed luc2/Neo, and no explanation in the figure legend. In many places, it is not mentioned whether the experiment was based on reporter or endogenous gene. Take Figure 2d as an example, it is not mentioned whether the data is from the endogenous promoter or the reporter promoter. The pulldown is the overexpressed YY1 or endogenous YY1? In Figure 2e-f, what are the YY1-a and YY1-b DNA fragments used in the gel shift assay? Only one copy of the binding site (a few nucleotides) or some promoter region containing the binding site? How long?

9. Figure 5b, it is not known what the data is. RNA-seq? qRT-PCR? The authors should try public databases or published RNA-seq data to analyze more control and patient lines.

10. Overall, the images of IF/FISH need better resolution. The puncta in neurites were not clear either.

Minor points:

1. The font size in figures is too small. Many places are barely readable.
2. Figure 2b: the boundary of the white bar is missing, not clear about the value.

REVIEWERS' COMMENTS

Reviewer #1 (Remarks to the Author):

An expansion of GGGGCC repeats within the C9ORF72 is the first genetic cause of Amyotrophic Lateral Sclerosis the 3rd most common neurodegenerative disease worldwide. How these repeats drive neuronal cell degeneration is still unclear. In this work, Chen and collaborator found that the YY1 transcription factor binds to and is titrated off its natural targets, notably binding to the Fuzzy gene promoter, ultimately resulting in mis-regulation of the Wnt/B-catenin pathway. Overall, Chen et al., findings are novel and provocative, relatively well controlled and well thought. However, this work raises several questions and require various additional experiments before to be considered for publication.

- Fig.1. How Fuzzy mis-regulation in C9 iPSC cells was found to stand out from other genes/candidates is unclear. RNA seq / screen or any other entry point showing Fuzzy mis-regulation as a solid candidate compared to other mis-regulations should be shown. Moreover, Fig. 1a/b show results and quantifications from classical RT-PCR and absolutely need to be confirmed by much quantitative assays (RNA Seq or real time qPCR).

Response: We appreciate the reviewer's insightful comments. *Fuzzy* was previously characterised as one of the ciliogenesis and planar polarity effector (CPLANE) genes. This group of genes also include *Inturned*, *Wdpcp*, *Rsg1*, and *Jbts17* (Toriyama *et al.*, Nature Genetics, 2016, **48**(6): 648–656). We performed qPCR on iPSC-derived spinal motor neuron samples to evaluate the expression of all CPLANE genes, *Fuzzy* was found to be the only one whose expression was dysregulated in disease neurons when compared to healthy control neurons (Fig. 1a). Gene expression analysis of the other CPLANE genes are included in the revised manuscript (Supplementary Fig. 1b-1e).

- Fig. 2. YY1 binding and activity on Fuzzy promoter and expression are relatively well controlled. However, binding of YY1 to GGGGCC RNA requires additional evidences, notably comparison with GGGGCC DNA and other RNA and DNA sequences, especially ones previously found to bind to YY1 as + controls. As negative controls, binding to randomized G4 C2 RNA mutants should be tested. Kd should be indicated.

Response: We thank the reviewer for his/her insightful comments. We have performed cell-free electrophoretic mobility shift assay to determine the binding affinities between YY1 and different DNA and RNA sequences (Supplementary Table 4). In particular, the AAV P5 DNA (Shi *et al.*, Cell, 1991, **67**(2): 377-88) and PP7 RNA (Wai *et al.*, Nucleic Acids Res, 2016, **44**(19): 9153-9165) were used as YY1-binding positive controls (Supplementary Fig. 3a, 4b). A randomised GGGGCC RNA was used as a scrambled control (Supplementary Fig. 4a). We also determined the binding affinities between YY1 and (GGGGCC)₈ RNA (Fig. 3a), *Fuzzy*^{YY1-a} DNA (Fig. 2a), *Fuzzy*^{YY1-b} DNA (Fig. 2a), and (GGGGCC)₈ DNA (Supplementary Fig. 4c). The binding affinities between YY1 and different sequences are summarised below. We found that the binding affinity of YY1–(GGGGCC)₈ RNA interaction (63.65 ± 8.79 nM) is much stronger than that with both *Fuzzy*^{YY1-a} DNA (237.33 ± 21.86

nM) and *Fuzzy*^{YY1-b} DNA (438.53 ± 74.40 nM). This finding further supports that GGGGCC RNA prevents YY1 from binding to the *Fuzzy* promoter.

YY1 binding partners	Binding affinities (K_a)
Fuzzy ^{YY1-a} DNA	237.33 ± 21.86 nM
Fuzzy ^{YY1-b} DNA	438.53 ± 74.40 nM
AAV P5 DNA	141.73 ± 14.58 nM
(GGGGCC) ₈ DNA	230.77 ± 5.40 nM
(GGGGCC) ₈ RNA	63.65 ± 8.79 nM
Randomised GGGGCC RNA	225.93 ± 40.11 nM
PP7 RNA	89.84 ± 8.52 nM

- Fig. 3. YY1 co-localization within GGGGCC RNA foci should be confirmed with another antibody or with Fluorescent Protein (FP)-tagged YY1. Titration/ sequestration and/or at least decreased mobility of FP-YY1 need to be tested by FRAP analysis and/or Dendra conversion (PS, an image of co-localization is not evidence of immobilization or sequestration, namely YY1 could be enriched within GGGGCC RNA foci but still able to move in and out). Eventually, comparison with a positive control (hnRNPH or hnRNPF) could be presented. Similarly, a negative control including a RNAase treatment would be a nice addition.

Response: We greatly appreciate the reviewer for pointing out these issues. We performed fluorescence *in situ* hybridisation coupled with immunocytochemistry and confirmed the co-localisation between endogenous hnRNP H protein and GGGGCC RNA foci in both our SK-N-MC transfected cell (Fig. 3b) and iPSC-derived spinal motor neuron models (Supplementary Fig. 9). Under the same experimental settings, we observed the co-localisation between endogenous YY1 protein and GGGGCC RNA foci (Fig. 3c, 5a; Supplementary Fig. 8). Such co-localisation was further confirmed using another anti-YY1 antibody (22156-1-AP, Proteintech) raised against a different epitope (Fig. 3d). We also demonstrated such co-localisation using an overexpressed fluorescent protein-tagged YY1 protein (Fig. 4b). Further, we showed that the co-localisation between YY1/hnRNP H protein and GGGGCC RNA foci was abolished upon RNase treatment (Fig. 3b-d, 5a; Supplementary Fig. 8, 9). Moreover, we performed the fluorescence recovery after photobleaching (FRAP) analysis, and found a lack of fluorescence recovery of YY1-mCherry (Fig. 3f) and hnRNP H-mCherry (Fig. 3g) after photobleaching in cells with GGGGCC RNA foci formation. This finding highlights the reduced mobility of YY1-mCherry and hnRNP H-mCherry proteins in mutant GGGGCC RNA cells.

IPS / motor neurons cell experiments sound well performed and well controlled, but as non-expert in such models, other Referee opinions will be required.

Response: We thank the reviewer for his/her comments. We performed additional experiments on iPSC-derived spinal motor neurons following the comments from another Referee. These experiments include:

1. The chromatin immunoprecipitation assay was performed to determine the binding of YY1 to *Fuzzy* promoter. We found that the binding of endogenous YY1 protein to the endogenous *Fuzzy*^{YY1} sites was perturbed in C9ALS/FTD spinal motor neurons. Overexpression of YY1 in patient neurons rescued such defect (Fig. 6a).
2. We performed nascent RNA capture assay to detect the level of nascent *Fuzzy* transcript. When compared to isogenic control spinal motor neurons, nascent *Fuzzy* transcript level was reduced in disease neurons. Such reduction was restored upon the overexpression of YY1 (Fig. 6b).
3. We included additional control and patient iPSC lines (Supplementary Table 1). All the control and patient lines were differentiated into spinal motor neurons. These cells were used for 1) the detection of *Fuzzy*/Fuzzy downregulation (Fig. 1a, 1b); 2) co-localisation study between YY1/hnRNP H and GGGGCC RNA foci (Fig. 5a, 5b; Supplementary Fig. 8, 9); and 3) study of Wnt/ β -catenin pathway target gene expression (Fig. 7d; Supplementary Fig. 11a, 11b).

Reviewer #2 (Remarks to the Author):

A (G4C2)_n expansion in *C9orf72* is the most common genetic cause of amyotrophic lateral sclerosis (ALS) and frontotemporal dementia (FTD). Repeat RNA gain of toxicity is recognized as one of the primary disease mechanisms. In this manuscript by Chen et al, the authors reported that the zinc finger (ZF) transcription regulator YY1 binds to GGGGCC repeat RNA and this interaction compromises the binding of YY1 to the *Fuzzy* promoter site, resulting in downregulation of *Fuzzy*. This consequently activates the Wnt/ β -catenin pathway, which induces synaptic deficits in C9ALS/FTD neurons. This study proposes a novel mechanism of repeat RNA mediated toxicity. However, more evidence is needed to fully support the proposed model.

Major concerns:

1. As the GGGGCC RNA can interfere with the binding of YY1 on both *Fuzzy* YY1-a and YY1-b regions, even bigger effect on 1-b which mediates the suppression of the promoter, why the RNA leads to downregulation of the targets predominantly? In Figure 2h, why repeat expression has no effect on the YY1-a mutant?

Response: We thank the reviewer for his/her insightful comments. YY1 functions as transcriptional activator and repressor at *Fuzzy*^{YY1-a} and *Fuzzy*^{YY1-b} sites, respectively (Fig. 2c). Mutation of *Fuzzy*^{YY1-a} site caused a dramatic reduction (~50-fold) of *Fuzzy* promoter activity. On the contrary, only a mild induction (~2.5-fold) of *Fuzzy* promoter activity was detected when *Fuzzy*^{YY1-b} site was mutated (Fig. 2c). Our data thus demonstrate that the disruption of YY1–*Fuzzy*^{YY1-a} interaction greatly inhibits *Fuzzy* promoter activity, whereas disruption of YY1–*Fuzzy*^{YY1-b} interaction only slightly induces *Fuzzy* promoter activity.

Moreover, we showed that stepwise overexpression of mutant GGGGCC RNA caused a progressive disruption of YY1 binding to both *Fuzzy*^{YY1-a} and *Fuzzy*^{YY1-b} sites (Fig. 2f). Interestingly, a steady increase in mutant GGGGCC RNA expression levels also led to a gradual inhibition of *Fuzzy*

promoter activity (Fig. 2g). This suggests that mutant GGGGCC RNA-mediated downregulation of *Fuzzy* transcription is mainly attributable to the impairment of YY1–*Fuzzy*^{YY1-a} interaction, which translates into a pronounced reduction of *Fuzzy* promoter activity (Fig. 2c). Under the influence of *Fuzzy*^{YY1-a}, the mild induction effect mediated through the disruption of YY1–*Fuzzy*^{YY1-b} interaction would likely be overshadowed.

Mutant GGGGCC RNA downregulates *Fuzzy* promoter activity through perturbing YY1 binding to wild-type *Fuzzy*^{YY1-a} site (Fig. 2c, 2f, 2g). Mutation introduced to the conserved nucleotide sequence in the YY1 binding site is known to disrupt YY1 binding to DNA (Riquet *et al.*, J Biol Chem, 2001, **276**(42): 38665-38672). Given the above, the mutant *Fuzzy*^{YY1-a} site (Fig. 1g) has already lost its ability to interact with YY1, thus mutant GGGGCC RNA expression has no effect on it (Fig. 1h).

2. In Figure 2b, YY1 knockdown should be performed to show the effect is mediated through YY1.

Response: We agree with the reviewer on this point. We found that when *YY1* expression was knocked down, *Fuzzy* promoter activity was inhibited (Fig. 2d). This inhibitory effect is similar to what we observed in mutant GGGGCC RNA-expressing cells (Fig. 1f). We thus hypothesise that the effect of mutant GGGGCC RNA on *Fuzzy* promoter activity is mediated through disrupting YY1–*Fuzzy* promoter interaction. Our electrophoretic mobility shift (Fig. 2e), ChIP (Fig. 2f) and luciferase (Fig. 2g) assay results all support this hypothesis.

3. The binding of YY1 on the repeat RNA needs more convincing evidence. In Figure 3a, the colocalization of YY1 with RNA foci is not obvious. The staining of YY1 in GGGGCC66 cells has no significant difference from the GGGGCC2 cells. And in all the YY1 staining data including reporter cells and iPSC-motor neurons, only one image of cells was shown. There is no quantification or statistical analysis of the differences. In Figure 3b, the difference between GGGGCC2 and GGGGCC66 is surprisingly small. This is hard to explain the specific effect on the function of YY1 by the GGGGCC66. The low amount of binding will not sequester sufficient proteins to influence the overall function.

Response: We thank the reviewer for raising his/her concerns. We used hnRNP H, a known GGGGCC RNA binding protein (Lee *et al.*, Cell Rep, 2013, **5**(5): 1178–1186), as a positive control to optimize our experimental settings. In both our transfected SK-N-MC cell model (Fig. 3b) and iPSC-derived spinal motor neurons (Supplementary Fig. 9), we detected the co-localisation between endogenous hnRNP H protein and GGGGCC RNA foci. Under the same experimental settings, we also detected the co-localisation between endogenous YY1 protein and GGGGCC RNA foci (Fig. 3c, 3d, 5a; Supplementary Fig. 8). The quantifications of the percentage of cells/neurons that show co-localisation between YY1 and GGGGCC RNA foci are now included in the revised manuscript (Fig. 3e, 5b). We further performed cell-free electrophoretic mobility shift assay to show that there is a direct binding between recombinant YY1 protein and GGGGCC RNA (Fig. 3a). These data provide more evidence in support of the binding between YY1 and GGGGCC RNA.

4. In Figure 3c,d, as the colocalization assay cannot show convincing data, an alternative method is needed to show the binding and the domain required for the binding. The gel shift assay could help.

Response: We appreciate the reviewer for raising his/her concerns. The recombinant full-length and deletion mutant YY1 proteins were generated. Their binding affinity with GGGGCC RNA were determined by cell-free electrophoretic mobility shift assay. The results are summarised below. Our data clearly show that YY1^{full-length} protein binds directly to GGGGCC RNA (Fig. 3a), and YY1^{ZF3} plays an important role in YY1–GGGGCC RNA binding (Fig. 4a).

(GGGGCC)₈ RNA binding partners	Binding affinities (K_a)
YY1 ^{full-length}	63.65 ± 8.79 nM
YY1 ^{ZF1+ZF2+ZF3}	56.36 ± 8.36 nM
YY1 ^{ZF1+ZF2}	106.52 ± 5.39 nM

5. Figure 3e, the two C9 lines are not consistent. And again there is no statistics. Only two C9 iPSC lines were used in the whole manuscript. The number is low considering the variation of individual human lines.

Response: We appreciate the reviewer for raising this issue. In the revised manuscript, we included additional control and patient iPSC lines (Supplementary Table 1) and differentiated them into spinal motor neurons for validation. The following experiments were further performed, and they include:

1. qRT-PCR and immunoblotting analysis to determine *Fuzzy* downregulation at transcript (Fig. 1a) and protein (Fig. 1b) levels;
2. fluorescence *in situ* hybridisation followed by immunocytochemistry to determine the co-localisation between endogenous YY1 protein and GGGGCC RNA foci (Fig. 5a, 5b; Supplementary Fig. 8); and
3. immunoblotting analysis to determine the upregulation of Wnt/β-catenin pathway targets (Fig. 7d; Supplementary Fig. 11a, 11b).

6. The authors should perform ChIP for the endogenous promoter in iPSC-differentiated motor neurons comparing control and C9 lines. And the transcription activity of the endogenous YY1 target should be measured (such as by Click-iT Nascent RNA capture) to show the transcription is changed in patient neurons. The same assays should be performed in cells overexpressing YY1 to show the rescue at the molecular signatures besides synaptic phenotypes.

Response: We greatly appreciate the reviewer for raising these issues. The ChIP assay was performed on iPSC-derived spinal motor neurons to examine the binding between endogenous YY1 protein and endogenous *Fuzzy* gene promoter. When compared to the isogenic control neurons, the binding of YY1 to *Fuzzy*^{YY1-a} and *Fuzzy*^{YY1-b} were impaired in disease neurons. Such impairment was alleviated upon YY1 overexpression (Fig. 6a). We further performed Click-iT nascent RNA capture assay to measure the level of nascent *Fuzzy* transcript in iPSC-derived spinal motor neuron samples. We found that nascent *Fuzzy* mRNA level was significantly reduced in patient neurons. When YY1 was overexpressed, such reduction was rescued (Fig. 6b). These findings suggest that YY1 overexpression ameliorates the molecular defects in C9ALS/FTD spinal motor neurons.

7. In Figure 4e, the GGGGCC repeat RNA levels should be measured to confirm the RNA expression is not altered due to cotransfection. This will exclude the possibility that the reduced cell death is due to the decreased repeat RNA level.

Response: We thank the reviewer for pointing this out. We determined GGGGCC RNA level in (GGGGCC)_{2/66}-expressing cells upon YY1 co-transfection. A pair of primers targeting the upstream *C9orf72* sequence of *pAG3-(GGGGCC)_{2/66}* constructs (Fig. 1c) was used to amplify the GGGGCC RNA. We found that the GGGGCC RNA level was not altered when YY1 was co-expressed (Supplementary Fig. 10b, c). This excludes the possibility that the suppression of cell death by YY1 overexpression was due to the decreased GGGGCC RNA level.

8. Many important details are missing in the writing. It is hard to read and get the information how each experiment was done. For example, how the luciferase data is normalized? In the method, it is mentioned dual luciferase. But the labeling in the figures showed luc2/Neo, and no explanation in the figure legend. In many places, it is not mentioned whether the experiment was based on reporter or endogenous gene. Take Figure 2d as an example, it is not mentioned whether the data is from the endogenous promoter or the reporter promoter. The pulldown is the overexpressed YY1 or endogenous YY1? In Figure 2e-f, what are the YY1-a and YY1-b DNA fragments used in the gel shift assay? Only one copy of the binding site (a few nucleotides) or some promoter region containing the binding site? How long?

Response: We thank the reviewer for those remarks. These experimental details are now presented in red font in the “Methods” and “Figure legends” sections in the revised manuscript. In particular, how the luciferase data is normalized is now explained from lines 679 to 681, 796 to 801, and 830 to 833 in the revised manuscript. The experimental details on ChIP assay are also included from lines 812 to 814 in the revised manuscript, and from lines 58 to 61 in the Supplementary materials. The experimental details on cell-free electrophoretic mobility shift assay are now included from lines 806 to 812 in the revised manuscript.

9. Figure 5b, it is not known what the data is. RNA-seq? qRT-PCR? The authors should try public databases or published RNA-seq data to analyze more control and patient lines.

Response: We thank the reviewer for pointing this out. The data in Fig. 7b and 7c (Fig. 5b in the original manuscript) are qRT-PCR results. This information is now included in both the figure and the revised manuscript from lines 558 to 561. We also analysed more control and C9ALS/FTD patient iPSC-derived spinal motor neuron samples from published RNA-seq datasets (GSE143743, GSE168831, NeuroLINCS Consortium, and AnswerALS data portal). The upregulation of *CCND1*, *FOSL1* and *PITX2* transcript levels in patient neurons could also be detected in these RNA-seq datasets (Table 1).

10. Overall, the images of IF/FISH need better resolution. The puncta in neurites were not clear either.

Response: Thank you for pointing this out. Images with higher resolution are shown in the revised manuscript.

Minor points:

1. The font size in figures is too small. Many places are barely readable.

Response: We thank the reviewer for raising this concern. The font size in the figures has been adjusted to improve the readability.

2. Figure 2b: the boundary of the white bar is missing, not clear about the value.

Response: We thank the reviewer for pointing this out. This figure is now revised to show the boundary of the white bar (Fig. 2c).

REVIEWER COMMENTS

Reviewer #1 (Remarks to the Author):

The authors adequately responded to my comments, and this work is now suitable for publication.

Reviewer #2 (Remarks to the Author):

This manuscript has been significantly improved. The authors have added more data to support the conclusions and modified the writing to provide more details. One remaining concern is the gel shift assay on RNA targets. Although the free RNA signal was gradually reduced with increasing amount of protein, the shifted complex bands were barely visible. The DNA gel shift data is very clear, but not the RNA gel. This raised concern on the quantification accuracy. And sometimes, what was shown on the gel seem not consistent with the quantification. The label on the YY1-RNA complexes covers a wide range, including the strong signals in the well. Take Fig 4A as an example, how was the protein-RNA complex quantified? Does the quantification also include the signals in the well or close to the well? It was mentioned "YY1-DNA (should it be RNA?) complexes were formed in the wells of the gel, indicating that they are large complexes and/or low in solubility". But it is hard to tell whether the signal in the well is due to the complex formation. The gel shift assay and its quantification could be improved to better support the direct binding between RNA and YY1.

Fig.6a, the colors of genes vs columns are confusing.

REVIEWER COMMENTS

Reviewer #2 (Remarks to the Author):

This manuscript has been significantly improved. The authors have added more data to support the conclusions and modified the writing to provide more details. One remaining concern is the gel shift assay on RNA targets. Although the free RNA signal was gradually reduced with increasing amount of protein, the shifted complex bands were barely visible. The DNA gel shift data is very clear, but not the RNA gel. This raised concern on the quantification accuracy. And sometimes, what was shown on the gel seem not consistent with the quantification. The label on the YY1-RNA complexes covers a wide range, including the strong signals in the well. Take Fig 4A as an example, how was the protein-RNA complex quantified? Does the quantification also include the signals in the well or close to the well? It was mentioned “YY1-DNA (should it be RNA?) complexes were formed in the wells of the gel, indicating that they are large complexes and/or low in solubility”. But it is hard to tell whether the signal in the well is due to the complex formation. The gel shift assay and its quantification could be improved to better support the direct binding between RNA and YY1.

Response: We thank the reviewer for raising his/her concerns. In the revised manuscript, we have adjusted the RNA gel shift images to better demonstrate the shifted YY1-RNA complex bands (Fig. 3a, 4a; Supplementary Fig. 4a, 4b). The ImageJ software was used to quantify the intensities of the free RNA bands and the shifted YY1-RNA complex bands that are resolved in the gel. These experimental details are also described in the revised manuscript between lines 713 to 718. In addition to gel shift assay, we further performed fluorescence polarisation (FP) assay to examine the binding between YY1 protein and GGGGCC RNA (Fig. 3b). The binding affinity obtained from FP assay ($K_d = 45.30 \pm 3.67$ nM) is similar to what was obtained from the gel shift assay ($K_d = 46.76 \pm 3.42$ nM). Both our gel shift and FP assay results support the direct binding between GGGGCC RNA and YY1 protein.

Fig.6a, the colors of genes vs columns are confusing.

Response: We thank the reviewer for pointing this out. The Fig. 6a has been modified to improve the clarity.

REVIEWERS' COMMENTS

Reviewer #2 (Remarks to the Author):

The authors have adequately addressed the concerns. This manuscript is suitable for publication.